# N-Formimidoylation/-iminoacetylation modification in aminoglycosides requires FAD-dependent and ligand-protein NOS bridge dual chemistry

Yung-Lin Wang[1], Chin-Yuan Chang[2], Ning-Shian Hsu [1], I-Wen Lo[1], Kuan-Hung Lin[1], Chun-Liang Chen[1], Chi-Fon Chang[1], Zhe-Chong Wang[1], Yasushi Ogasawara [3], Tohru Dairi [3], Chitose Maruyama[4,5], Yoshimitsu Hamano [4,5] ✉ & Tsung-Lin Li [1,6] ✉

Oxidized cysteine residues are highly reactive and can form functional covalent conjugates, of which the allosteric redox switch formed by the lysine-cysteine NOS bridge is an example. Here, we report a noncanonical FAD-dependent enzyme Orf1 that adds a glycine-derived N-formimidoyl group to glycinothricin to form the antibiotic BD-12. X-ray crystallography was used to investigate this complex enzymatic process, which showed Orf1 has two substrate-binding sites that sit 13.5 Å apart unlike canonical FAD-dependent oxidoreductases. One site could accommodate glycine and the other glycinothricin or glycylthricin. Moreover, an intermediate-enzyme adduct with a NOS-covalent linkage was observed in the later site, where it acts as a two-scissile-bond linkage facilitating nucleophilic addition and cofactor-free decarboxylation. The chain length of nucleophilic acceptors vies with bond cleavage sites at either N−O or O−S accounting for N-formimidoylation or N-iminoacetylation. The resultant product is no longer sensitive to aminoglycoside-modifying enzymes, a strategy that antibiotic-producing species employ to counter drug resistance in competing species.

ST-F the first known congener out of the streptothricins (STs) family was isolated from the culture of *S. lavendulae* in 1943 (Fig. 1a)[1]. Its chemical framework is composed of an unusual aminosugar, carbamoylated *D*-gulosamine, to which two nonproteinogenic amino acids, streptolidine (a bicyclic streptolidine lactam) and L-β-lysine (β-Lys), are adorned respectively at the anomeric carbon and C2-amine. Analogs with various lengths of β-Lys oligopeptides such as ST-D and ST-E (with oligo(β-Lys)$_3$ and oligo(β-Lys)$_2$, respectively.), which exhibit relatively higher antibacterial activity[2]. Other ST congeners with structural nuances exist, such as BD-12 (Fig. 1a, compound **2**) which has an unusual N-formimidoylglycine in place of the β-Lys oligopeptide[3–5], and glycinothricin (Fig. 1a, compound **1**), which is similar to BD-12 but lacks the N-formimidoyl group (Fig. 1a)[6]. STs are broad-spectrum antimicrobials that inhibit protein synthesis in bacteria, fungi, and insects, but not phytogrowth[7–14]. However, the severe cytotoxicity of STs, such as nephrotoxicity, limits their clinical use[15–18]. To reduce STs' toxicity in

[1]Genomics Research Center, Academia Sinica, Taipei 11529, Taiwan. [2]Department of Biology Science and Technology, National Yang Ming Chiao Tung University, Hsinchu 30010, Taiwan. [3]Graduate School of Engineering, Hokkaido University, Kita-ku, Sapporo, Hokkaido 060-8628, Japan. [4]Graduate School of Bioscience and Biotechnology, Fukui Prefectural University, Eiheiji-cho, Fukui 910-1195, Japan. [5]Fukui Bioincubation Center (FBIC), Fukui Prefectural University, Eiheiji-cho, Fukui 910-1195, Japan. [6]Biotechnology Center, National Chung Hsing University, Taichung City 402, Taiwan. ✉e-mail: hamano@fpu.ac.jp; tlli@gate.sinica.edu.tw

eukaryotes, substantial efforts with organic chemistry-based strategies were made but remained in vain[19].

The biosynthesis of STs from various *Streptomyces* species has been extensively elucidated[20–22]. Of them, the biosynthesis of BD-12 (accession no. LC122485) differs from others as it recruits a Fem-like enzyme to install a glycine at the C2-amine of streptothrisamine in a tRNA-dependent manner, resulting in the production of glycinothricin[23,24].

Orf1, which encodes a *N*-formimidoyl fortimicin A synthase-like enzyme, has been proposed to convert glycinothricin to BD-12. In silico analysis indicates that Orf1 is a flavoprotein oxidoreductase (monooxygenase, FPMO or oxidase, FPOX). FPMO or PFOX are classified into three main categories (I-III), which are further subdivided into nine subgroups (A-I)[25,26]. Orf1 belongs to category III since its own substrate reduces flavin as opposed to those requiring an external NAD(P)H (category I;

**Fig. 1 | Chemical structures of STs and ST-related antibiotics. a** The gene product of *orf1* encoded in the BD-12 producing strain is able to catalyze glycinothricin to BD-12. Other than glycinothricin, glycylthricin, 3-aminopropionylthricin, 4-aminobutylthricin and ST-F can serve as substrates of Orf1 to bring on *N*-formimidoylated or *N*-iminoacetylated corresponding products. **b** The *N*-formimidoylation or *N*-iminoacetatylation was presumed to follow multiple steps of reactions in one single reaction site as does a typical FAD-dependent enzyme. **c** The proposed mechanisms of *N*-formimidoylation or *N*-iminoacetylation catalyzed by Orf1 that recruits a FAD prosthetic group and an NOS protein-ligand covalent modification to enable multiple reactions to take place at two separate but adjacent reaction chambers. Four-membered peroxide and three-membered oxaziridine that may respectively react with oxidized and non-oxidized cysteine to form the NOS linkage, however, were not detected in any circumstances. In the presence of proper acceptors (e.g., **1** or **4**), products **2**, **5** with the *N*-formimidoyl modification are formed by Orf1, in which sulfenic C281 is recyclable (reactivated only once). R'HN' colored blue is glycylthricin analogs (e.g., compound **7**) with various aliphatic side chains in length leading to *N*-iminoacetyl modification. In this circumstance, apart from hydrogen peroxide ($H_2O_2$) sulfenic C281 can be regenerated by water as determined by isotope labeling experiments using $H_2^{18}O$ highlighted red (see Fig. 5c).

the one-component) or a NAD(P)H-dependent flavin reductase (category II; the two-component)[26]. The *N*-formimidoyl group in *N*-formimidoyl fortimicin A originates from glycine through multiple reactions, including glycine oxidation, amine addition, second oxidation, and decarboxylation (Fig. 1b). The enzyme (Fsm14) that catalyzes this modification is characterized as a FAD-dependent oxidase (FPOX) because two stoichiometric quantities of hydrogen peroxide ($H_2O_2$) are produced at the expense of one equivalent of glycine[27]. However, this conclusion conflicts with the current understanding of FAD-dependent oxidative decarboxylation, where $H_2O_2$ commonly engages with a presumed imino product of the first-half reaction. Furthermore, it remains unclear how multiple reactions are hierarchically executed in a single enzyme (Fig. 1b)[28–30].

Bacterial resistance to aminoglycoside antibiotics that inhibit protein biogenesis can occur by decreasing antibiotic uptake/accumulation (e.g. efflux pumps), altering 16 S rRNA, other ribosomal proteins, or modifying chemical structures of antibiotics per se (e.g. aminoglycoside acetyltransferases, AACs)[31]. The inactivation of STs has been ascribed largely to chemical modification of the oligo(β-Lys) sidechain by β-*N*-acetylation (Fig. 1a) as manifested in the ST producers *S. lavendulae*[32,33], *S. rochei*[34], and *S. noursei*[35], where a universal gene encoding an *N*-acetyltransferase (NAT) is applied to the acetylation modification at the β-amine group of the β-Lys pharmacophore. Whether the *N*-formimidoyl modification is immune or succumbs to *N*-acetylation by ST-NAT alike (e.g. SttE from *S. lavendulae*) is still an unanswered issue. An encouraging example is that FDA-approved β-lactam antibiotic imipenem that is *N*-formimidoylated thienamycin[36], demonstrating superior efficacy to the latter for both drug resistance and stability. Similarly, *N*-formimidoyl fortimicin A (dactimicin) is insusceptible to AAC3 as well as has lower nephrotoxicity[37,38].

Here we report that Orf1 is a FAD-dependent enzyme encompassing two separate reaction centers one catalyzing formation of a reactive aldoxime-like intermediate and the other catalyzing a *N*-formimidoyl substitution reaction. Together, these reactions make Orf1 an noncanonical FAD-dependent monooxygenase (functionally Orf1 should be termed as *N*-formimidyol/iminoacetyl synthase) (Fig. 1c). Most amazingly, the enzyme evolves an unprecedented N−O−S enzyme-substrate linkage rod mechanism to mediate nucleophilic addition and FAD-peroxide-independent decarboxylation reactions. The *N*-formimidoyl modification is a result of molecular evolution enriching the structural landscape of STs, intrinsically diversifying the types of molecules and leveraging bacterial drug-resistance.

## Results and discussion
### In vitro/in vivo characterization of Orf1
BD-12 features an unusual *N*-formimidoyl functional group at the terminus of its glycyl sidechain in contrast to a typical ST that displays primary amine groups at its oligo(β-Lys)$_n$ sidechain. Previously, we identified the BD-12 biosynthetic gene cluster (BGC) in *Streptomyces luteocolor* NBRC 13826 (Supplementary Fig. 1) and heterologous host strains (*Streptomyces lividans* TK23 and *Streptomyces avermitilis* SUKA17) expressing the BGC that produces BD-12[(ref24)]. It has been known that the glycyl moiety is transferred to streptothrisamine by a

Fem-like enzyme (*orf11*) at the expense of one molecule of Gly-tRNA to form **1** prior to the addition of the *N*-formimidoyl group[24]. The *orf1* gene that encodes a *N*-formimidoyl fortimicin A synthase-like enzyme in the BD-12 BGC was thought to commit the formimidoylation reaction in the presence of glycine[27]. To confirm this hypothesis, the *orf1* gene was inactivated by an in-frame deletion and introduced into *S. avermitilis* SUKA17. The resulting transformant, SUKA17-BD-12-Δorf1, was cultured and analyzed by high-performance liquid chromatography and high-resolution electrospray ionization mass spectrometry (HPLC-HR-ESI-MS). Compared to TK23-BD-12, carrying the BD-12 biosynthetic gene cluster (wild type), the deletion of *orf1* abolished the biosynthesis of BD-12 (Supplementary Fig. 1b). In a further HPLC-HR-ESI-MS analysis of the culture broth, the TK23-BD-12-Δorf1 strain was shown to produce a compound (m/z 460.215 [M + H]$^+$) rather than BD-12 (m/z 487.226 [M + H]$^+$). The mass decrease by 27 Da was in good agreement with the fact that the *orf1* gene homologous to an *N*-formimidoyl synthase gene was inactivated; the Δ27 Da compound seemed to be **1**. To confirm the chemical structure of **1**, it was purified from the culture broth, and NMR analysis clarified that compound **1** is indeed glycinothricin (Supplementary Fig. 2; Supplementary Table 1). Moreover, these results strongly suggested that **1** was the biosynthetic precursor of BD-12 and a substrate of the *orf1* gene product (*N*-formimidoyl synthase).

Two glycylation reactions executed by two different types of enzymes in a row are rare if any. The *N*-formimidoyl modification is peculiarly intriguing as a barrage of reactions apparently goes with the territory starting with free glycine oxidation and ending on a bulky recipient with a unique *N*-formimidoyl substitution. To explore this fascinating issue, recombinant Orf1 expressed in *E. coli* displays yellow tetrameric proteins in the solution while eluting out of gel filtration chromatography (Supplementary Fig. 3a, b). The yellow color indicates each polypeptide chain harbors a flavin prosthetic group as opposed to the formylglycine-generating enzyme (FGE), which is a copper I dependent enzyme[39]. This yellow cofactor was determined to be flavin adenine dinucleotide (FAD) by subjecting the supernatant of denatured proteins to LC-MS analysis (Supplementary Fig. 3c). Biochemical assays were performed in parallel, wherein reaction mixtures containing glycine and a chemically synthesized glycylthricin (Fig. 1a, compound **4**) instead of **1** were quenched in due course and subjected to LC analysis. When compared with the control (addition with denatured Orf1), a new peak that emerged on the LC trace matched with the synthetic standard in the same retention time, mass unit and fragmentation pattern, suggesting that Orf1 is a flavin-dependent enzyme capable of *N*-formimidoylating the terminal glycyl sidechain of glycylthricin to **5** in the presence of glycine. To confirm that the *N*-formimidoyl moiety is derived from an intact molecule of glycine, we performed Orf1-mediated reactions with the addition of glycine, ($^{15}N$)-glycine, ($1–^{13}C$)-glycine, ($2–^{13}C$)-glycine, ($2–^{13}C$ $^{15}N$)-glycine, or ($1,2–^{13}C_2$–$^{15}N$)-glycine alongside **4**. LC-MS spectra clearly demonstrate that the *N*-formimidoyl group is derived from one intact molecule of glycine by an increment of 1 or 2 Da respectively from ($^{15}N$)- or ($2–^{13}C$ $^{15}N$)-glycine but no change from ($1–^{13}C$)-glycine (Fig. 2a, b). In the EIC (selected at m/z 459.2) trace a small peak emerged other than **5** (Fig. 2a), indicating an *N*-iminoacetyl-glycylthricin

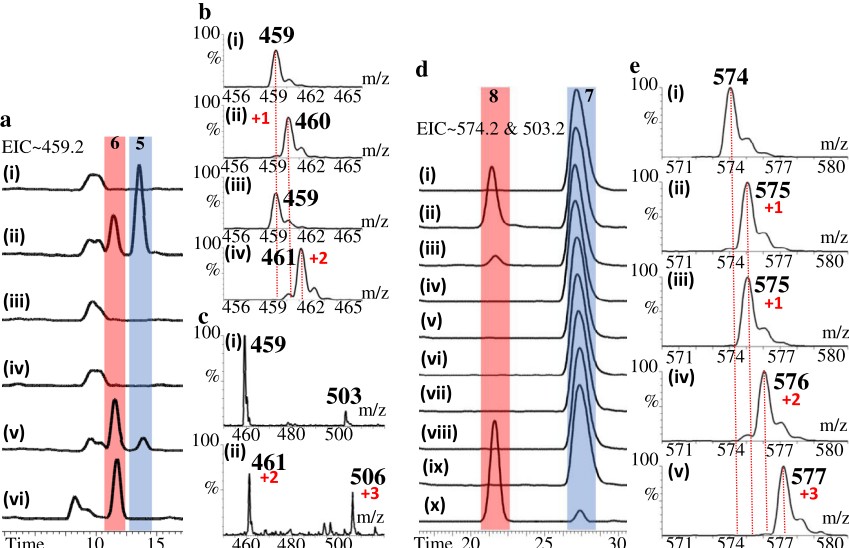

**Fig. 2 | The biochemical examinations for Orf1 and mutants thereof against glycylthricin and ST-F. a** The EIC traces of formimidoyl-glycylthricin (Fig. 1a, compound **5**) (m/z 459.2 [M + H]+) for reactions catalyzed by (i) denatured Orf1, (ii) Orf1, (iii) C281A, (iv) C281S, (v) E312A, or (vi) F316A. **b** Mass spectra of **5** produced in the Orf1-mediated reactions in the presence of **4** and (i) glycine ([M + H]+ = 459), (ii) 15N-glycine ([M + H]+ = 460), (iii) 1-13C-glycine ([M + H]+ = 459), or (iv) 2-13C-15N-glycine ([M + H]+ = 461). **c** Mass spectra of iminoacetyl-glycylthricin (Fig. 1a, compound **6**) produced in the Orf1-mediated reactions in the presence of **4** and (i) glycine ([M + H]+ = 503), or (ii) 13C2-15N-glycine ([M + H]+ = 506). **d** The EIC traces of ST-F (Fig. 1a, compound **7**) (m/z 503.2 [M + H]+) and iminoacetyl-ST-F (Fig. 1a, compound **8**) (m/z 574.2 [M + H]+) for reactions catalyzed by (i) denatured Orf1, (ii) Orf1, (iii) Orf1 with addition of DTT, (iv) C281S, (v) R342A, (vi) C281S and R342A, (vii) ThiO and R342A, (viii) E426Q, (ix) E312A, or (x) F316A. **e** Mass spectra of the Orf1-mediated reactions in the presence of ST-F and (i) glycine ([M + H]+ = 574), (ii) 15N-glycine ([M + H]+ = 575), (iii) 1-13C-glycine ([M + H]+ = 575), (iv) 2-13C-15N-glycine ([M + H]+ = 576), or (v) 13C2-15N-glycine ([M + H]+ = 577). Source data are provided as a Source Data file.

(Fig. 1a, compound **6**) as shown by gaining 44 Da of a carboxyl group. The **6** was further confirmed given an increment of 3 Da in the presence of (1,2–13C2–15N)-glycine, where the moiety of **5** in the spectra suggested it is a daughter ionic fragment of **6** (Fig. 2c). The enzymatic reactions in the presence of ST-F (Fig. 1a, compound **7**) alongside glycine or isotopically labeled glycine suggested that the product is *N*-iminoacetyl-ST-F (Fig. 1a, compound **8**) (Fig. 2d, e). NMR analysis of **8** purified from the reaction mixture revealed that the *N*-iminoacetyl group was attached to the ε-amino group of the β-lysine moiety (Supplementary Fig. 4; Supplementary Table 2).

The kinetics for a single substrate (glycine; the first phase of the overall reaction) and double substrates (glycine + ST-F) under the pseudo-first-order condition were estimated, on which the kinetic parameters are comparable for both the single substrate ($K_m$: 202 μM, $k_{cat}$: 0.737 min−1, $k_{cat}K_m^{-1}$: 0.00365 min−1μM−1) and the double substrate (WT: $K_m$: 196 μM, $k_{cat}$: 3.08 min−1, $k_{cat}K_m^{-1}$: 0.0157 min−1μM−1; F316A: $K_m$: 190 μM, $k_{cat}$: 7.16 min−1, $k_{cat}K_m^{-1}$: 0.0377 min−1μM−1) suggesting that the glycine oxidation phase is likely the rate-limiting step (Supplementary Figs. 5, 6).

To explore the substrate scope of Orf1, we assayed Orf1 against a number of clinically used aminoglycosides, including kanamycin, gentamycin, amikacin or sisomicin in the presence of glycine. Among these, only amikacin and sisomicin were found *N*-formimidoylation (Supplementary Figs. 7, 8). We subsequently assayed Orf1 versus synthetic 3-aminopropionylthricin (Fig. 1a, compound **9**), 4-aminobutylthricin (Fig. 1a, compound **10**) or **7** in the presence of glycine. Surprisingly, in addition to the *N*-formimidoyl modification, a new *N*-glycine imine akin to that in kasugamycin was found amid the selected acceptors[40], where the ratio of *N*-glycine imination versus *N*-formimidoylation is positively proportional to the length of the sidechain (Supplementary Fig. 9). The *N*-glycine imination is unexpected and perplexing as it is somewhat in conflict with the general FAD-mediated decarboxylation reactions (see below). Both the *N*-formimidoylation and -glycine imination are useful biotransformations concerning diversity-oriented diversification/complexification, not least

a strategy to counter aminoglycoside-modifying enzymes (e.g. aminoglycoside-acetylating enzyme, AAC) to avoid drug resistance (see below).

## Crystal structures and substrate-binding sites

Regarding the glycine-derived formimidoyl transfer reaction, it must proceed through a multitude of chemical reactions as mentioned above. Our instinct tells us that the enzyme should be similar to a typical FAD-dependent monooxygenase, where the oxidized imino acid intermediate should undergo a nucleophilic attack of FAD-associated peroxide, thus resulting in an oxidative decarboxylation consequence. However, an issue that arises right away is when the addition and transfer reaction take place and how it works out in sequence at the same reaction site. To crack the reaction order and corresponding mechanisms, we took advantage of X-ray crystallography, whereby snapshots of complexes in phases may help illustrate the overall reaction continuums. At the protein level, Orf1 is a flavoprotein similar to glycine oxidases ThiO and GO (both are members of the glutathione reductase 2 (GR2) family of proteins)[41,42] containing two domains: one for FAD binding and the other for substrate binding (the Dali Z-score and r.m.s.d. are 41.7 and 2.4 Å (359 Cα atoms)[43], respectively). Orf1 is a protein of 491 amino acid residues longer than ThiO or GO by an additional 100 or so residues at the sequence level, which is folded into 12 α-helices and 17 β-strands into a three-dimensional (3D) structure (Fig. 3a, b). It has a quaternary structure of four subunits (tetrameric) in solution. The 3D structure of Orf1 in the presence of glycine was crystallized and determined at 1.9 Å as illustrated in Fig. 3a, wherein an asymmetric unit contains 8 polypeptide chains as a result of the low-symmetry P1 point group. The quaternary structure of Orf1 was confirmed by PDBePISA with an immense tetrameric buried area (11390 Å2)[44]. Each individual polypeptide chain is folded into three structural domains, a FAD binding domain, a substrate binding domain and a twisted 4-helix bundle domain (additional 100 C-terminal residues in consistency with sequence analysis), which has no known function with no equivalents

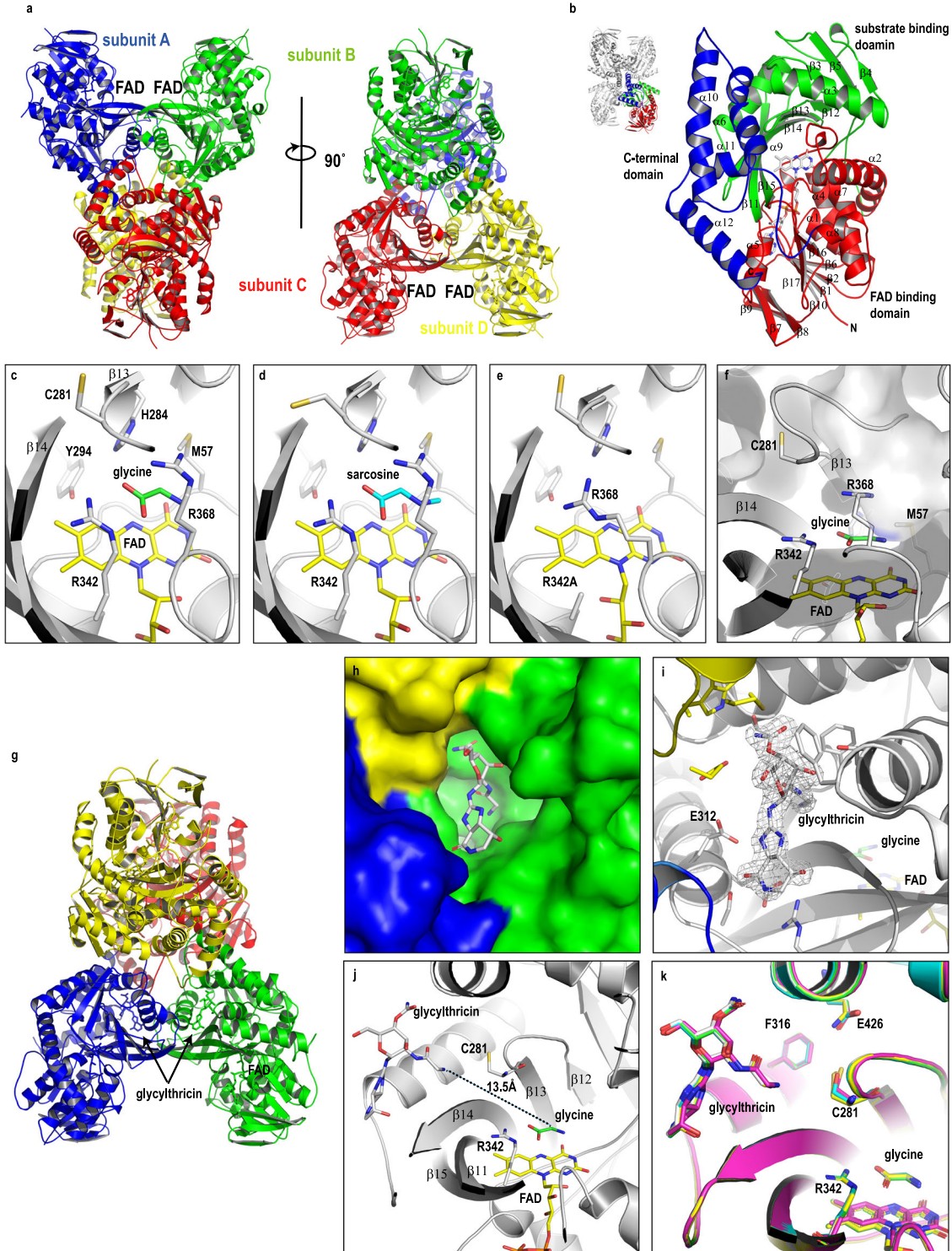

**Fig. 3 | Overall structures of Orf1. a** Orf1 is a tetrameric complex of a dimer of dimers in solution, of which each protomer contains a FAD prosthetic group. **b** Each protomer is comprised of three domains, a FAD binding domain (red), a substrate binding domain (green) and a C-terminal domain (blue). The tetrameric complex is interfaced by the substrate binding domain and the C-terminal domain. **c** The glycine binding site is located on the top of the *re* face of the isoalloxazine ring of FAD and shaped by residues A54, G55, A56, M57, Y294, H284, R342, and R368 revealed from the Orf1-glycine complex. **d** Sarcosine also binds to the glycine-binding site as revealed from the Orf1-sacrosine complex. **e** R368 is misplaced when R342 was mutated to Ala resulting in the loss of enzyme activity as revealed from the Orf1-R342A structure. **f** The surface presentation of the glycine binding pocket, where a cavity nearby the active site was speculated the glycylthricin binding site. **g** The complex structure of Orf1-glycine-glycylthricin. Each Orf1 protomer in the

tetrameric complex contains one molecule of FAD, glycine and glycylthricin. **h** The glycylthricin binding site is located at the inter-subunit junctions, where binding pockets are formed. **i** The 2$F_o$-$F_c$ density map of glycylthricin is countered at 1 σ shown in a gray mesh. **j** The distance between the amine nitrogen atom of glycyl-thricin and the α-carbon of glycine is 13.5 Å away from each other. The antiparallel-β-sheet (β11- β15) acts as a physical barrier that delimits a boundary for both the glycylthricin and glycine binding pockets. **k** Superposition of crystal structures of wildtype and mutants in complex with glycylthricin. The Orf1-glycylthricin (B chain), C281S-glycylthricin (B chain), F316A-glycylthricin (B chain), R342A-glycylthricin (A chain) and E426Q-glycylthricin (B chain) are colored green, cyan, gray, magentas and yellow, respectively. The major binding-site residues and gly-cylthricin are well aligned without significant conformational changes.

in all GR2 family proteins reported thus far (see below, Fig. 3b), presumably taking part in protein-protein interaction at the quaternary structure level.

Each single structure orchestrates two apparent binding cavities one for FAD and the other one for glycine. The FAD prosthetic cofactor in Orf1 is akin to that in other GR2 subfamily proteins, where the binding site is distant from the monomer-monomer interfaces. It adopts an extended conformation with the isoalloxazine ring compartmentalizing the FAD-binding domain and the glycine-binding domain that sits above the *re*-face of the isoalloxazine ring (Fig. 3c–f).

The compartment divided by the FAD cofactor is enclosed by four splayed β-strands curving around the isoalloxazine ring with two loop regions that define the glycine binding site (Fig. 3f). The crystals treated with glycine or sarcosine showed an extra electron density where the α-C atom of glycine or sarcosine is best described as a sp³ configuration. Except for G55 and H284, the residues A54, M57, R368 and R342 that surround the glycine/FAD reaction center are conserved between Orf1 and ThiO in the secondary and ternary sequence level (Supplementary Fig. 10). The glycine is anchored in place by: (a) $N^{η2}$ and $N^ε$ of R342 that H-bond with the terminal carboxylate of glycine (2.8 and 2.9 Å) and π-stack with the guanidine group of R368; (b) the α-C atom of glycine is at a H-bonding distance to the N5 atom of the isoalloxazine ring (3.0 Å); (c) the amine group of glycine forms a H-bond with the carbonyl group of R368 (2.9 Å). H284 and Y294 are associated with each other via H-bonding (4.1 Å) in close proximity to the amino group (4.9 Å) of glycine, presumably serving as a catalytic dyad to deprotonate α-amine of glycine in support of the hydride transfer mechanism[28,44], as the activity of site-directed mutants H284A and Y294F was compromised to some extent. Moreover, R342 is another conserved residue in GO (glycine oxidase), DAAO (*D*-amino acids oxidase) and SOX (sarcosine oxidase) that electrostatically interacts with the COO⁻ group of substrates[42,45,46].

An appreciable volume of a cavity (539 Å³) close to the FAD/glycine binding site was initially considered the glycylthricin binding site, as it appears accessible through a narrow entrance for the glycyl side chain of glycylthricin to reach the oxidized glycine (Fig. 3f; Supplementary Fig. 11a). The true binding site, however, is located at the tetrameric inter-junction entwined by the C-terminal four-bundle domains on the opposite side of the one misdeemed. This site will not be known until the determination of the glycylthricin-glycine-Orf1 ternary complex structure (Fig. 3g; Supplementary Fig. 12), where it is composed of two neighboring domains with a tangible volume of 1415 Å³ (Fig. 3h). Glycylthricin resides nicely inside the cavity lined by interacting residues (F279, R306, S310, E313, E312, T423, E434, W435, M438, T466 and F470). Specifically, the carbamoylated *D*-gulosamine and streptolidine lactam moieties are held by residues E434, T423, T466 and E312 through H-bonding networks (Fig. 3i; Supplementary Fig. 13). The side chains of E312 and E434 likely experience significant conformational changes upon substrate binding (Supplementary Fig. 13). Mutant E312A confirmed the necessity of changes in recognizing substrates, whereby the binding affinity versus ST-F dropped 10-fold, in addition to losing the reactivity to form **8** (Fig. 2d (ix); Supplementary Table 3). Both glycylthricin and 4-aminobutylthricin (**10**, a product analog) were observed in the same cavity in structures, suggesting that it is an intrinsic binding site for glycylthricin (Supplementary Fig. 12a, b). However, the glycine amine of glycylthricin is distant from the α-C atom of glycine in the glycine binding site as far as 13.5 Å (Fig. 3j). Geographically, the antiparallel-β-sheet (β11- β15) of the glycine binding domain serves as a physical barrier, separating glycylthricin from glycine. Such an arrangement negates multiple reactions in one site but instead supports two separate but adjacent reaction centers. Namely, the reactive product generated from glycine oxidation in the FAD-mediated

reaction is safeguarded and swiftly transferred to the glycylthricin binding site, allowing subsequent addition, decarboxylation, dehydration reactions to proceed towards the *N*-formimidoylated or *N*-iminoacetylated modification in order.

## Identification of a covalently-modified adduct at C281

In the Orf1-glycine complex structure, an extra chunk of electron density extended from the terminal thiol group of C281, which was unexpectedly spotted (the only residue out of 12 cysteine residues in a single polypeptide chain) at the glycylthricin binding site, a location next door to the glycine binding cavity and 14.7 Å away from the N5 of FAD (Fig. 3j). This electron density appears to exhibit two different lobes a short one and an extended one (at the "H subunit" of the structure) (Fig. 4a). For the latter, it best fits with a *trans* glycine imine-N–O–S-C281 species based on the density contour, a structure reminiscent of the NOS bridge recently reported in transaldolases (*Ng*TAL), where a covalent crosslink between cysteine and lysine residues was determined as an allosteric redox switch (see below, Fig. 4b, c)[47]. We found that the geometric properties of the glycine imine NOS bridge (N-S distance 2.64 Å, NOS angle 119.9°, N-O distance 1.4 Å and O-S distance 1.65 Å) fully comply with the criteria defined by Tittmann et al. for a NOS bridge in proteins[48]. The C281 residue, nonetheless, is in a reduced form in all polypeptide chains in the structures added with no glycine. It is worth noting that each data set collected from crystal diffraction of X-ray has an accumulated average absorbed dose of 0.67 MGy, which is far lower than the dose needed to provoke radiation damage thereby suggesting that the C281-adduct is not a result of X-ray radiation but rather an intrinsic NOS covalent adduct. We reasoned that the thiol group is subject to reactive oxygen species (ROS, e.g. $H_2O_2$)-mediated cysteine oxidation to form sulfenic acid (Fig. 1c), a relatively short-lived transient species, which is pivotal to its transformation to other oxidative post-translational modifications (e.g., sulfinic acid (Fig. 5a) and disulfide bond)[49].

To confirm this, Orf1 was incubated with glycine, digested with trypsin, treated with 5,5-dimethyl-1,3-cyclodehexadione (a reagent used to detect the appearance of sulfenic acid) and then subjected to mass spectrometry analysis[50–52]. The protein ID unequivocally displayed that the peptide mass is increased by 138 a.m.u. because of a sulfur-dimedone adduct, thus confirming the existence of sulfenic acid. The C281 adducts in the form of a sulfinic modification (+33 Da) were also detected by MALDI-TOF (Fig. 5a) in a manner similar to all the ramifications in C38 oxidation observed for the K8A variant of *Ng*TAL (the enzyme where the lysine-cysteine switch was uncovered)[47].

To probe the importance of C281, we mutated it to alanine or serine and E426 to Q426 (a residue related to NOS-formation). Biochemical assays confirmed that the both residues and the NOS bridge are essential for the reaction to proceed, as neither one led to the end product (Fig. 2a (iii, iv), d (iv, viii)). In contrast, both C281A and E426Q mutants still exhibited similar glycine oxidation activity, and the R342A lost the glycine oxidation ability but held the same ST-F binding affinity as the wild type (Supplementary Table 3). In terms of conformations, the mutants showed no difference from the wild type except for the sidechain of R368 in the R342A mutant (Fig. 3e, k). Residue R342 turned out to be critical because the sidechain of R368 was misplaced by intruding into the glycine binding site as manifested in the crystal structure of R342A (Fig. 3e). Next, we performed coupled assays to detect the end product to see if it could be formed by complementary pairs C281S/R342A or ThiO/R342A (Fig. 2d (vi, vii)). No product was found, suggesting that (1) glycine imine (the product of ThiO) is not the intermediate for R342A (with intact C281); (2) the first-reaction-phase product formed by C281S is water-labile and unable to transfer undamagedly from C281S to R342A. Given that **4** is in close proximity to the C281-S–O–N-glycine imine adduct (see below), the reactions, such as addition, decarboxylation, and dehydration, are likely to take place in sequence towards the formation of *N*-formimidoyl-glycylthricin. We

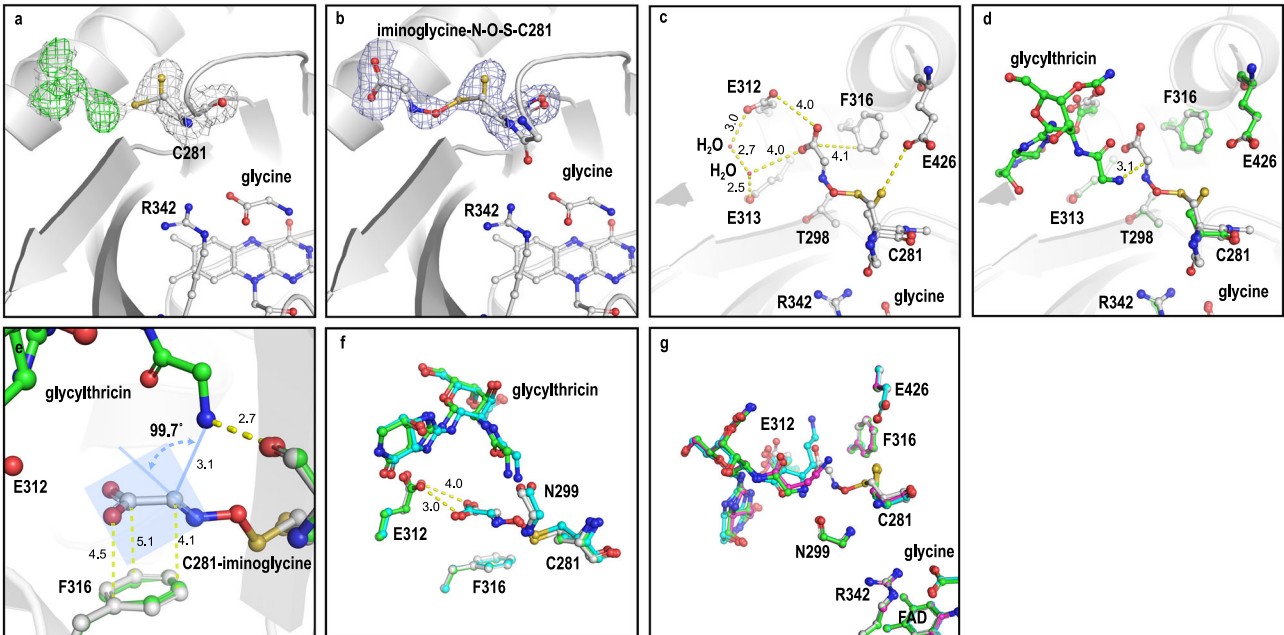

**Fig. 4 | The glycine imine-N−O−S-C281 bridge in the Orf1-glycine complex structure. a** An intrinsic electron density (contrasted by $2F_o$-$F_c$ map contoured at 1 σ in gray and the $F_o$-$F_c$ map contoured at 3 σ in green) was unexpectedly found nearby residue C281 at the glycylthricin binding site. **b** The chemical model that best fits into the electron density region ($2F_o$-$F_c$ map at 1 σ colored blue) is a glycine imine adduct in a covalent linkage to C281 through a N−O−S bridge. **c** The C281-iminoglycine adduct is surrounded by residues T298, E312, E313, F316 and E426. The distance between two atoms was labeled with a yellow dashed line. **d**, **e** Superposition of two complexes (the C281-adduct and the glycylthricin-containing structures) shows that the distance between the amine group of glycylthricin and α-carbon of the iminoacetate adduct is 3.1 Å within a general H-bond range with a Burgi-Dunitz angle of 99.7° in an approaching trajectory. **f** Superposition of WT-glycylthricin (green), E312A-glycylthricin (cyan) and C281-iminoglycine adduct (gray) reveals that E312 undergoes a substrate-induced conformational change, in which upon addition the resulting decarboxylation facilitates β-elimination. **g** Superposition of WT-glycylthricin (green), WT-4-aminobutylthricin (magenta), F316-ST-F (cyan) and C281-iminoglycine adduct (gray) shows considerable displacements of the median γ-aminobutyl and the long β-lysine sidechains away from that of glycylthricin likely as a result of a geographic effect.

also observed that 90% of the enzyme activity was lost when reactions were conducted in a DTT-containing (100 mM) reaction solution, likely because glycine oxidation cannot proceed and/or C281 is fully reduced (underscoring the essentiality of the post-translational modification) (Supplementary Table 3). The residues nearby the C281-adduct include T298, E312, E313, F316 and E426 (Fig. 3c), of which F316, E312 and E313 may cooperatively allow nucleophilic addition and cofactor-free decarboxylation reactions to proceed in sequence. Our reasoning is that F316 stabilizes the planar imine glycine adduct through the π-π stacking interaction. The importance of this residue was ascertained by mutational experiments, whereby F316A brings on **6** as the major product (Fig. 2a (vi)). Overlaying two complexes (the C281-adduct and the glycylthricin-containing structures) reveals that the glycylthricin ligand is positioned by residues E312 and E313 in place, making the terminal amine group of glycylthricin in close proximity to the α-carbon of the oximinoacetate adduct by 3.1 Å with a Burgi-Dunitz angle of 99.7° an approach trajectory (Fig. 4d, e). The nucleophilicity of the terminal amine may be enhanced after deprotonation by the main chain carbonyl group of N299 (2.7 Å). The attack of the amine group from the *re* face onto the F316-stabilized C281-adduct should reconfigure the planar (sp$^2$) *trans* imine to a tetrahedral (sp$^3$, *S*-configuration) geminal diamine species, whereby the π-π stabilization no longer holds. The charged carboxyl group is neutralized after the entropy-driven decarboxylation reaction ensuing in response to the repulsion force incited between the charged and hydrophobic groups. The left electron lone pair could flow into the σ-antibonding orbital of the nitrogen atom of the anti-periplanar N−O−S bridge in a way akin to an E2-like elimination reaction, which differs to the canonical PLP or thiamine-dependent (both cofactors serving as an electron sink) decarboxylation reactions. E312 another key residue collaborates with F316 to facilitate the decarboxylative reaction by introducing a repulsive electrostatic force. The fact is that the residue experiences a substrate-induced conformational change bringing the terminal carboxylate group in a short distance (3.3 Å) to the carboxylate group of the oximinoacetate moiety (Fig. 4f; Supplementary Fig. 13) evidenced by mutational experiments whereby compound **6** becomes the major product in the presence of E312A (Fig. 2a (v)). As a result, E312 undergoes a substrate-induced conformational change, in which upon addition the induced decarboxylation should facilitate β-elimination. On the other hand, superposition of the F316A-ST-F and C281-adduct complex displays that the long side chain of ST-F is subject to a geographic effect (Fig. 4g), resulting in flipping of the amide group away from E312 to accommodate the entire aliphatic sidechain. This posture is somewhat unexpected as the ε-amine of the β-lysine is a bit extended away from the α-carbon of the oximinoacetate adduct likely due to the absence of the phenyl ring (F316A). E312 not only recognizes the streptolidine base but also ushers the β-lysine to crawl into the binding site in a compact manner. In this context, E312A is able to catalyze the formation of **5** and **6** but no more **8**, as opposed to F316A, which catalyzes the formation of **8**, the dominant product (Fig. 2a (v), d (ix, x)).

## Catalytic mechanisms

Since the glycine imine is not the intermediate based on the coupled assays (ThiO/R342A and C281S/R342A), a glycine imine-like reactive intermediate produced at the FAD-reaction center should act on the relatively electrophilic sulfur of C281-SOH to give rise to the glycine imine-N−O−S-C281 adduct. This reactive species is not trappable because of water lability. This reactive species formed at the first reaction center is tunneled to the second one, where it is placed with its carboxylic group in association with the Glu426-water dyad. One tunnel that possibly channels the oxidized glycine into the

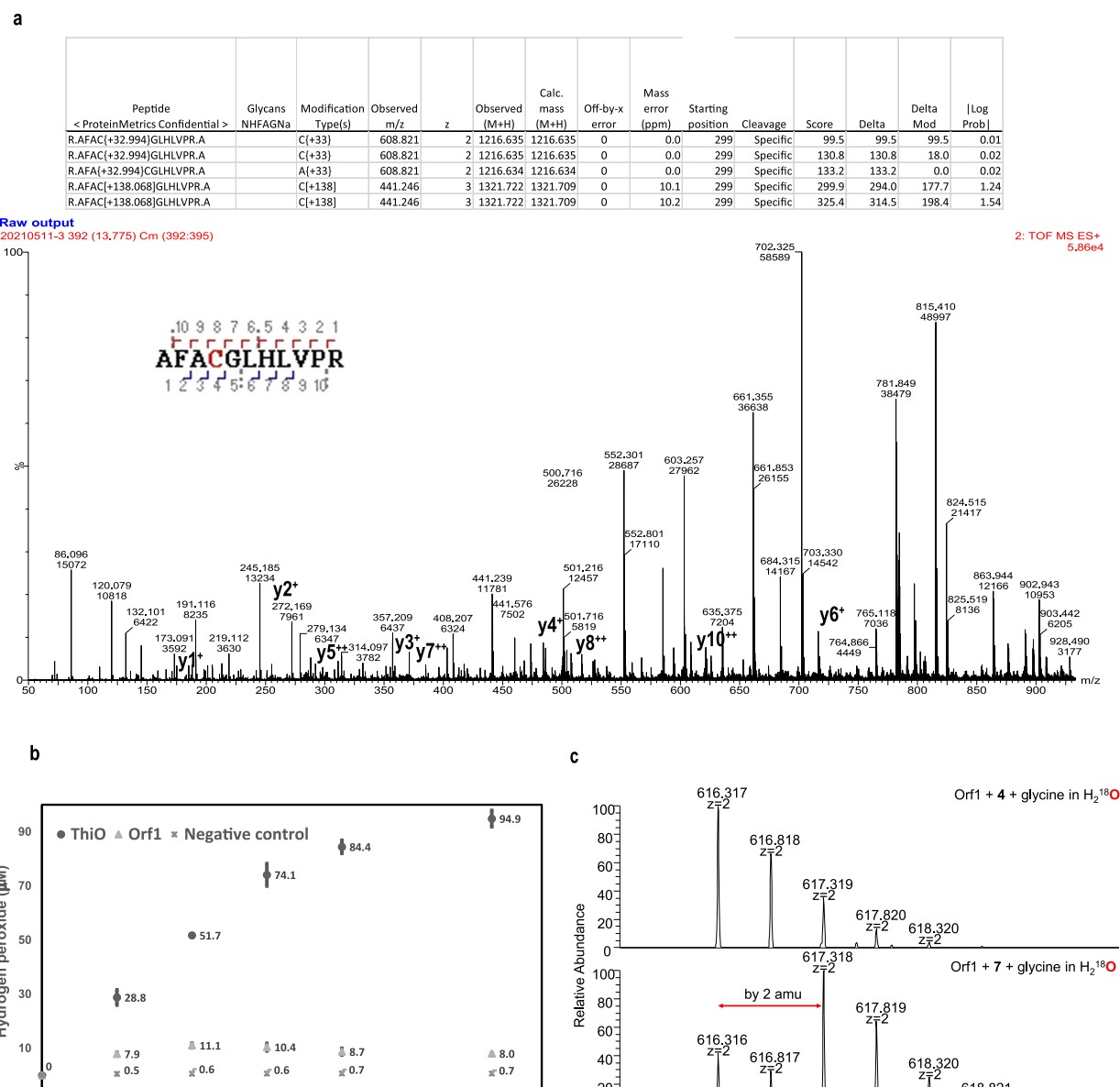

**Fig. 5 | The determination of covalent modifications of residue C281. a** Mass spectrometric analyses for the C281-dimedone adduct and the sulfinylated peptide fragments of Orf1 incubated in a glycine-containing buffer solution. Detection of the AFAC281GLHLVPR peptide labeled with dimedone (+138 Da) and the sulfinylated peptide (+33 Da). MS/MS spectrum indicates the formation of a C281-dimedone adduct confirming the existence of the SOH group. The mass spectrometric analysis concurrently revealed that the thiol can undergo stepwise oxidation to sulfenic, sulfinic and sulfonic acid (Supplementary Fig. 29). The conversion rate for SH-Cys281 to SOH-Cys281 is fast and high. The mass spectrometry files were deposited in ProteomXchange with identifier PXD041105 and PXD041104, respectively. **b** A considerable amount of $H_2O_2$ detected in the glycine-containing reaction solution added with ThiO (FPOX) in 3 min in contrast to much less $H_2O_2$ in the same context but added with Orf1 (FPMO). The assays of Orf1 and ThiO were run with two independent replicates ($n = 2$) and the data were presented as mean values ± SD.

Source data are provided as a Source Data file. Photospectrometric analysis further supports Orf1 a FPMO (Supplementary Fig. 30). **c** The sulfonic-containing peptides (at C281) from trypsin-treated Orf1 were selected and analyzed by mass spectrometry, in that the sulfonic-containing fragment renders relatively high abundance with molecular distinctiveness. Orf1 was incubated in the presence of **4** or **7** in a $H_2^{18}O$ buffer solution 1 h and then subjected to trypsin digestion overnight, of which the mass spectra demonstrated that the incorporation of one $^{18}O$-isotopic atom in sulfonic-containing fragments occurs mainly in the reaction with **7** (lower panel, with an enriched M + 2 peak) as opposed to that with **4** (upper panel, with a typical isotopic profile) in favor of the mechanism proposed in Fig. 1c. z = 2 stands for doubly-charged positive ions in mass spectrometry. The reaction condition is the same as that for the experiments shown in Supplementary Fig. 31c(ii) and Supplementary Fig. 9e(ii). Two major products are **5** and **8**, respectively. The mass spectrometry files were deposited in ProteomXchange with identifier PXD041103.

*N*-formimidoylation site was located using CAVER 3.0 there are four predicted tunnels around the FAD/glycine reaction center one for the oxidized glycine channeling and the other three for entrance of glycine, (Supplementary Fig. 11a)[53]. Five residues that constitute the tunnel and would block the intermediate tunneling if mutated were individually made (L124D, L124E, A278V, A280S and H419Y). Except for insoluble L124E, all the mutants L124D, A278V and H419Y lost the *N*-iminoacetylation activity unable to form **8**, while they are still structurally sound and active for glycine oxidation (Supplementary Fig. 11b, c). The intermediate should be analogous to *N*-OH-glycine imine or equivalents (e.g., glycine epoxide (carboxyl oxaziridine), or cyclic glycine peroxide) (Fig. 1c). This given setting aligns the

## Table 1 | Data collection and refinement statistics

| pdb code | Orf1-Apo 7XYE | Orf1-glycine 7XQA | Orf1-sarcosine 7XXD |
|---|---|---|---|
| Data collection | P1 | P1 | P1 |
| Space group | | | |
| Cell dimensions | | | |
| $a, b, c$ (Å) | 103.7, 108.5, 134.9 | 103.4, 107.8, 134.8 | 103.9, 108.9, 134.7 |
| $\alpha, \beta, \gamma$ (°) | 90.1, 90, 83.8 | 89.9, 90, 83.6 | 90.02, 90, 83.3 |
| Resolution (Å) | 30–2.47 (2.56–2.47)[a] | 30–1.93 (2.00–1.93) | 30–1.98 (2.05–1.98) |
| $R_{merge}$ | 5.8 (41.0) | 4.3 (23.2) | 5.9 (79.3) |
| $I / \sigma I$ | 23.4 (3.2) | 35.8 (5.9) | 25.6 (2.1) |
| Completeness (%) | 97.5 (93.0) | 97.9 (96.5) | 98.0 (96.8) |
| Redundancy | 4.4 (4.3) | 5.3 (5.3) | 4.6 (4.3) |
| **Refinement** | | | |
| Resolution (Å) | 29.7–2.48 (2.55–2.48) | 29.7–1.93 (1.98–1.93) | 28.7–1.98 (2.03–1.98) |
| No. reflections | 195367 | 424882 | 379436 |
| $R_{work}/R_{free}$ | 22.0, 24.7 | 19.2, 20.8 | 21.2, 23.5 |
| No. atoms | 30359 | 32835 | 32439 |
| Protein | 29090 | 29239 | 29208 |
| Ligand/ion | 424 | 474 | 472 |
| Water | 845 | 3122 | 2759 |
| $B$-factors | 41.7 | 23.6 | 32.7 |
| Protein | 45.2 | 25.0 | 34.8 |
| Ligand/ion | 41.3 | 22.3 | 35.4 |
| Water | 32.8 | 30.5 | 39.2 |
| R.m.s. deviations | | | |
| Bond lengths (Å) | 0.007 | 0.007 | 0.007 |
| Bond angles (°) | 1.46 | 1.37 | 1.42 |

| pdb code | Orf1-glycine-4 7XXC | Orf1-glycine-10 7XXM | C281S-glycine-4 7XX0 |
|---|---|---|---|
| Data collection | P1 | P1 | P1 |
| Space group | | | |
| Cell dimensions | | | |
| $a, b, c$ (Å) | 103.5, 108.1, 134.6 | 103.4, 107.2, 135.2 | 103.7, 108.1, 135.8 |
| $\alpha, \beta, \gamma$ (°) | 89.8, 90.1, 96.5 | 90.1, 90.04, 83.7 | 90.1, 90, 96.6 |
| Resolution (Å) | 30–1.99 (2.06–1.99) | 30–2.05 (2.12–2.05) | 30–2.20 (2.28–2.20) |
| $R_{merge}$ | 3.7 (25.6) | 4.7 (60.0) | 5.9 (52.3) |
| $I / \sigma I$ | 38.1 (5.8) | 30 (2.5) | 24.9 (3.6) |
| Completeness (%) | 97.5 (93.2) | 98.2 (97.1) | 97.5 (96.3) |
| Redundancy | 5.2 (5.2) | 4.6 (4.5) | 4.6 (4.4) |
| **Refinement** | | | |
| Resolution (Å) | 27.8–1.99 (2.04–1.99) | 29.4–2.12 (2.17–2.12) | 29.9–2.20 (2.26–2.20) |
| No. reflections | 387913 | 302700 | 289233 |
| $R_{work}/R_{free}$ | 19.8, 22.1 | 19.5, 21.5 | 18.4, 20.6 |
| No. atoms | 32355 | 32019 | 32003 |
| Protein | 29225 | 28875 | 29215 |
| Ligand/ion | 704 | 720 | 704 |
| Water | 2426 | 2424 | 2084 |
| $B$-factors | 29.4 | 31.0 | 33.8 |
| Protein | 32.2 | 32.8 | 36.2 |
| Ligand/ion | 27.7 | 47.7 | 36.0 |
| Water | 33.7 | 35.3 | 38.5 |
| R.m.s. deviations | | | |

## Table 1 (continued) | Data collection and refinement statistics

| pdb code | Orf1-glycine-4 7XXC | Orf1-glycine-10 7XXM | C281S-glycine-4 7XX0 |
|---|---|---|---|
| Bond lengths (Å) | 0.007 | 0.007 | 0.007 |
| Bond angles (°) | 1.42 | 1.41 | 1.41 |

| pdb code | R342A-4 7XXR | E426Q-glycine-4 7XYL | F316A-glycine-7 7XXP |
|---|---|---|---|
| Data collection | P1 | P1 | P1 |
| Space group | | | |
| Cell dimensions | | | |
| $a, b, c$ (Å) | 103.4, 107.9, 133.4 | 103.4, 107.4, 135 | 103.3, 106.4, 135.5 |
| $\alpha, \beta, \gamma$ (°) | 90.1, 90, 96.3 | 89.98, 89.99,83.6 | 90, 90.01, 96.2 |
| Resolution (Å) | 30–2.40 (2.49–2.40) | 30–2.10 (2.18–2.10) | 30–2.05 (2.12–2.05) |
| $R_{merge}$ | 6.0 (64.2) | 5.4 (66.1) | 8.1 (61.7) |
| $I / \sigma I$ | 25.0 (2.7) | 29.6 (2.4) | 17.5 (2.4) |
| Completeness (%) | 98.5 (97.7) | 98.2 (97.1) | 97.5 (93.2) |
| Redundancy | 4.4 (4.3) | 4.9 (4.3) | 4.3 (3.9) |
| **Refinement** | | | |
| Resolution (Å) | 28.6–2.40 (2.46–2.40) | 28.1–2.10 (2.15–2.10) | 29.9–2.05 (2.04–1.99) |
| No. reflections | 212738 | 319322 | 347963 |
| $R_{work}/R_{free}$ | 20.0, 22.0 | 19.7, 22.3 | 18.8, 20.8 |
| No. atoms | 30395 | 32497 | 31707 |
| Protein | 29176 | 29212 | 29136 |
| Ligand/ion | 664 | 704 | 744 |
| Water | 555 | 2548 | 1827 |
| $B$-factors | 43.2 | 31.8 | 29.9 |
| Protein | 46.5 | 35.2 | 31.9 |
| Ligand/ion | 52.4 | 29.9 | 30.8 |
| Water | 32.9 | 38.9 | 32.1 |
| R.m.s. deviations | | | |
| Bond lengths (Å) | 0.008 | 0.007 | 0.008 |
| Bond angles (°) | 1.48 | 1.41 | 1.49 |

| pdb code | F316A-glycine-4 7YOX | E312A-glycine-4 8GRI |
|---|---|---|
| Data collection | P1 | P1 |
| Space group | | |
| Cell dimensions | | |
| $a, b, c$ (Å) | 103.6, 107.9, 134.8 | 103.3, 108.1, 133.5 |
| $\alpha, \beta, \gamma$ (°) | 89.96, 90.1, 83.4 | 89.8, 89.98, 96.4 |
| Resolution (Å) | 30–2.12 (2.20–2.12) | 30–2.37 (2.45–2.37) |
| $R_{merge}$ | 5.7 (60.8) | 8.6 (63.1) |
| $I / \sigma I$ | 28 (2.8) | 17.6 (2.4) |
| Completeness (%) | 97.4 (91.2) | 99.3 (99) |
| Redundancy | 4.2 (4.3) | 5.1 (5.1) |
| **Refinement** | | |
| Resolution (Å) | 27.7–2.05 (2.10–2.05) | 29.7–2.37 (2.43–2.37) |
| No. reflections | 352917 | 211837 |
| $R_{work}/R_{free}$ | 19.9, 21.5 | 20.9, 23.3 |
| No. atoms | 31217 | 30454 |
| Protein | 29118 | 29064 |
| Ligand/ion | 704 | 704 |
| Water | 1395 | 686 |

## Table 1 (continued) | Data collection and refinement statistics

| pdb code | F316A-glycine-4 7Y0X | E312A-glycine-4 8GRI |
|---|---|---|
| *B*-factors | 29.3 | 42.0 |
| Protein | 31.4 | 45.3 |
| Ligand/ion | 32.1 | 43.4 |
| Water | 30.5 | 33.5 |
| R.m.s. deviations | | |
| Bond lengths (Å) | 0.007 | 0.006 |
| Bond angles (°) | 1.44 | 1.37 |

| pdb code | SttE-CoA-4 7YPU | SttE-7 7YPV |
|---|---|---|
| **Data collection** | P212121 | P1 |
| Space group | | |
| Cell dimensions | | |
| *a, b, c* (Å) | 52.4, 102.8, 386.4 | 48.7, 77.8, 109 |
| α, β, γ (°) | 90, 90, 90 | 89.95, 90.1, 89.99 |
| Resolution (Å) | 30–2.36 (2.44–2.36) | 20–2.42 (2.51–2.42) |
| $R_{merge}$ | 5.8 (37.2) | 6.6 (43.9) |
| I / σI | 39.4 (6.5) | 15.2 (3.8) |
| Completeness (%) | 99.7 (98.9) | 97.9 (96.9) |
| Redundancy | 10.2 (8.9) | 2.8 (2.7) |
| **Refinement** | | |
| Resolution (Å) | 29.8–2.36 (2.42–2.36) | 19.9–2.42 (2.48–2.42) |
| No. reflections | 82980 | 56666 |
| $R_{work}/R_{free}$ | 19.7, 22.3 | 21.1, 24.5 |
| No. atoms | 10706 | 11119 |
| Protein | 9937 | 10302 |
| Ligand/ion | 450 | 280 |
| Water | 319 | 537 |
| *B*-factors | 33.3 | 34.4 |
| Protein | 35.0 | 37.2 |
| Ligand/ion | 61.4 | 30.1 |
| Water | 35.0 | 35.7 |
| R.m.s. deviations | | |
| Bond lengths (Å) | 0.006 | 0.005 |
| Bond angles (°) | 1.42 | 1.41 |

aValues in parentheses are for highest-resolution shell.

nucleophilic hydroxylamine towards the electrophilic sulfur atom of C281-SOH, resulting in the *trans* oximinoglycine with a N–O–S covalent linkage to the protein. The adduct is stabilized by the Glu426-water dyad through a low barrier H-bond (2.5 Å) and poised for a nucleophilic addition reaction with glycylthricin as described above. To our knowledge, the NOS-mediated decarboxylation is unorthodox to all known biological decarboxylation reactions, not least α-amino acids that typically require a pyridoxal-phosphate cofactor[30]. The protein-NOS-mediated decarboxylation reaction empirically follows the heterolytic ionic mechanism because no radical signal was detected by EPR as that of glycyl radical enzymes (Supplementary Fig. 3d). Notwithstanding other likelihoods, several oxime-like precedents do exist. For instance, the plant cytochrome P450 monooxygenase (Cyp79A1) in the biosynthesis of cyanogenic glucoside dhurrin is a close example[54,55]. Cyp79A1 catalyzes two consecutive *N*-hydroxylation reactions to yield *N,N*-(OH)₂-tyrosine, which then undergoes dehydration and tautomerization to α-nitrosocarboxylic acid. The resulting α-nitroso species serves as an electron sink and is protonated intermolecularly by the terminal carboxylic acid leading to (*E*)-*p*-hydroxyphenylacetaldoxime (retaining the N–O linkage) upon decarboxylation. In the present study, the N–O linkage of oximinoglycine is cleaved as a consequence of nonoxidative decarboxylation, whereby C281-SOH is pushed away in a good leaving group manner. On the other hand, given ST-F the reaction gives rise to *N*-iminoacetyl glycinthricin, where the terminal carboxylic group is retained and the N–O bond undergoes dehydration afterwards. This N–O–S bridge acts as though a linkage rod with two scissile bonds. The chain length of the nucleophile is another determinant, of which the stress/strain provoked in the binding site as well as the trajectory of the nucleophilic addition reaction together leverages the bond cleavage at either the nitroxide (N–O) or sulfur oxide (O–S) site (Fig. 1c).

### Orf1 counteracts SttE

Almost all ST BGCs encode a ST-resistant *N*-acetyl transferase (NAT), a self-resistance enzyme, that acts on STs. Given the *stt*E gene encoded in the *Streptomyces lavendulae* BCRC12163, the gene product was expressed and subjected to probe its drug resistance scope. A new major peak emerged on the LC trace in a reaction solution containing acetyl-CoA, glycylthricin and SttE when compared with control in which SttE was omitted or glycylthricin was replaced by *N*-formimidoyl glycylthricin (Supplementary Fig. 14). The mass-spectrometric analysis showed that the molecular weight of the peak is increased by 42 Da in the solution containing 4, suggesting that glycylthricin but not 5 is subject to *N*-acetylation. In the microbial susceptibility assay, the *E. coli* test strain that harbored a *stt*E-containing plasmid displayed a diminished inhibition zone, in which glycylthricin or ST-F was added, indicating that SttE is a ST-modifying enzyme and the *N*-fomimidyla-tion or *N*-iminoglycination is antagonistic towards the acetylation of SttE (Supplementary Fig. 15). The bifunctional aminoglycoside modifying enzyme AAC(6′)-Ie-APH(6″)-Ia inactivated glycylthricin by *N*-acetylation to some extent in in vitro assay (Supplementary Fig. 16)[56], while the *N*-formimidoylated glycylthricin was not sensitive to this given modifying enzyme. It is likely owing to the steric hindrance, reaction orientation as well as the reduced nucleophilicity of the sp² *N*-formimidoyl group of 5 where the double bond may resonate between two nitrogen atoms at the terminal of the side chain.

We then determined the structures of SttE in complex with ST-F or glycylthricin with CoA, of which crystals were packed in space group P1 and P212121 and determined at resolutions of 2.42 and 2.36 Å, respectively (Supplementary Fig. 17). The structure of SttE revealed a single core domain of a general control non-derepressible 5 (GCN5)-related *N*-acetyltransferase (GNAT) architecture, which is composed of seven extended β-strands and three α-helices. Both the extended strand β7 and β-hairpin between αA and β2 establish the substantial protein interface promoting protein dimerization, where D164 specifically interacts with R101 and W139 via electrostatic interactions. These binary complexes identified two ligand-binding sites where strands β3 and β4 splay out evolving a typical β-bulge motif. The AcCoA donor sits astride the fork with the acyl head facing the sidechain of the glycyl-thricin acceptor. The AcCoA binding site is composed of β4, αB, β5 and αC. Glycylthricin, on the other hand, is held by interacting residues (K75, F59, P60, D63, G54, W98, E105, D106, E141, T143, L167, Y168, S173 and E176), which as a whole define the ST binding site. In brief, residues D63, E141 interact with the guanidino group of streptolidine lactam by three H-bonds, while residues S176 and E176 associate the carbamoylated D-gulosamine with two H-bonds. The amine group is in close proximity to the sulfur atom of CoA in distances of 3.3–4.8 Å (c.f. SttE-glycylthricin vs. SttE-ST-F). D106 appears to be the base deprotonating the β-amine group for the acetylation modification to take place, while it is a bit far away from the terminal amine group of 4. Its nucleophilicity, none-theless, may well be enhanced by two mainchain oxo groups (E105 and E141), thus accounting for the equivalent activity of SttE towards 4 and 7 (Supplementary Fig. 17c).

In this study we report on a noncanonical FAD-dependent enzyme, Orf1, classified in the group-transferase protein family that

contains two adjacent but separate reaction centers. The first center runs a FAD-mediated oxidation-oxygenation dual reaction to give rise to a N-hydroxyl glycine imine-like product in a net 4-electron oxidation manner. The second center is 13.5 Å away from the first one, where residue C281 plays an indispensable role. To become a fully active enzyme, the sulfhydryl group of C281 undergoes a post-translational modification with ROS (e.g., leaking $H_2O_2$ from the FAD-dependent reaction) to sulfenic acid (Fig. 5b). The sulfenic group is capable of reacting with an oxygenated glycine imine species passed down from the first reaction center via a subtle one-way tunnel to form the NOS-covalent adduct. This enzyme-bound adduct then reacts with glycinothricin or glycylthricin following a barrage of chemical reactions, including nucleophilic addition, cofactor-free decarboxylation, β-elimination and dehydration, to bring on BD-12 (N-formimidoyl gly-cylthricin). This group-transfer modification is unprecedented as it seamlessly integrates two unrelated chemical strategies at two separate locations in one single enzyme by utilizing oligomerization to invent a second reaction center/substrate binding site. The resultant N-formimidoylation is no longer sensitive to AAC aminoglycoside-modifying enzymes, a biochemical strategy that the producing strain evolved to reinvigorate its armory to counter the orthodox drug-resistance mechanism leveraged by competing species. The substrate scope of the enzyme is not limited to the innate ones but is expandable at both ends of donors and acceptors, thereby befitting as a versatile biocatalyst in view of diversifying/complexifying privileged compounds. The NOS-mediated chemistry catalyzed by Orf1 should not be the sole case, the sequence fingerprint characterizing the activity associated with this type of chemistry may guide in finding more enzymes featuring the same mechanism (Fig. 1c).

## Methods

### Cloning and protein purification

The *orf1* gene was amplified from the BD-12 producing strain *Streptomyces luteocolor* ATCC29775[(ref24)]. The *orf1*-containing pET28a was transformed into the *Escherichia coli* BL21 (DE3) strain. One liter LB medium containing 50 mg l⁻¹ kanamycin was cultured at 37 °C until $A_{600}$ reaching 0.6 and induced with 0.2 mM isopropyl β-D-thiogalactopyranoside (IPTG). After 20 h of incubation at 16 °C, the cells were harvested by centrifugation and ruptured by sonication in 30 ml lysis buffer (20 mM phosphate pH 8.0, 500 mM NaCl, 20 mM imidazole and 5% (v/v) glycerol). Cell debris was span down by centrifugation (48000 g). Supernatants were loaded into a 5 ml $Ni^{2+}$-NTA column and washed with 50 ml lysis buffer. Yellow Orf1 was eluted with 25 ml elution buffer (20 mM phosphate pH 8.0, 500 mM NaCl, 250 mM imidazole and 5% (v/v) glycerol). Protein was further purified by size-exclusion chromatography with a Hiload 16/60 Superdex 200 pg column under an isocratic condition (20 mM Tris·HCl pH 7.5 and 150 ml NaCl), and its purity was estimated by 10% SDS-PAGE. Protein concentrations ware determined using NanoDrop. The primers for site-directed mutagenesis were designed by QuikChange (Agilent) and synthesized by Genomics (Taipei) (Supplementary Table 4). All mutants were confirmed by DNA sequencing, and corresponding proteins were purified using the same protocol for wild-type Orf1. The *thiO* gene was amplified from the *Escherichia coli* JM109 strain. The *sttE* gene was cloned from the *Streptomyces lavendulae* BCRC 12163 strain[20]. The *aac(6')-Ie-aph(2'')-Ia* gene[56] was synthesized by Genomics (Teipei). Both ThiO and SttE were expressed and purified using the same protocol as that for Orf1 (Supplementary Figs. 18, 19).

### Crystallization and data collection

Orf1 and its mutants were crystallized using the hanging-drop vapor diffusion method at 20 °C. The protein was concentrated to 15 mg ml⁻¹ in 25 mM HEPES pH 7.5 and mixed with the same volume of the reservoir solution (0.1 M Tris·HCl pH 8.5, 0.2 M KBr, 8% (w/v) PEG 550 MME and 8% (w/v) PEG 20000). After a week incubation, the complex crystals were prepared by soaking 10 mM substrates for 5 mins. The crystals were immersed in 30% (v/v) glycerol as the cryoprotectant and frozen with liquid nitrogen. All X-ray diffraction data were collected using ADSQ Quantum 315, MX300-HE or EIGER2 16 M detectors at 100 K on beamlines TLS-13B1, TLS-15A1, TPS-05A and TPS-07A at the National Synchrotron Radiation Research Center (NSRRC), Taiwan. The datasets were indexed and scaled to P1 space group using the HKL2000 package[57]. The diffraction-weighted dose for the C281-iminoglycine dataset was calculated using RADDOSE-3D (beam size ~200 nm, flux ~3.01E + 12 photon/sec and attenuation ~68.3%)[58]. The crystals of SttE were obtained from the 15 mg ml⁻¹ protein mixed with 3 mM ST-F or 3 mM glycylthricin and 3 mM coenzyme A in the reservoir solution (0.1 M MES pH 6.5, 0.2 M ammonium sulfate and 12.5% (w/v) PEG 8000).

### Structure determination and refinement

Orf1 and SttE structures were determined by using the molecular replacement method MOLREP[59]. The search model was generated from the FFAS03 server[60] using ThiO (PDB code 1NG3) and SttE's homolog (PDB code 3PP9) as the templates, respectively. Polypeptide models were built and refined with REFMAC5 and COOT[61,62]. Detailed refinement statistics are shown in Table 1. The omit maps were generated using PHENIX[63]. Structural figures were plotted using PyMOL.

### Mass spectrometry analysis of Orf1

To perform mass spectrometry analysis of the C281-dimedone adduct, 0.2 mM Orf1 was incubated with 1 mM glycine 37 °C for 10 mins and added with 10 mM dimedone at room temperature for another 30 mins. The reaction solution was exchanged to a buffer solution containing 25 mM $NH_4HCO_3$ with 40 µg µl⁻¹ trypsin. After overnight incubation at 37 °C, the sample (2 µg µl⁻¹) was injected at a flow rate of 50 µl ml⁻¹ into an ACQUITY UPLC CSH C18 peptide column (130 Å, 1 mm × 150 mm, 1.7 µm, #186006935) and detected by LC-ESI-MS, a Waters SELECT SERIES™ Cyclic IMS QToF (Manchester, UK). In brief, the gradient was set with 98% (v/v) buffer A for 3 mins to 60% (v/v) buffer B over 20 mins, where buffer A was water with 0.1% (v/v) formic acid and buffer B was acetonitrile with 0.1% (v/v) formic acid. The mass was operated in the LC/MS$^E$ mode of data acquisition, where alternating 1 second scans for low (4 V) or high (10-40 V) collision energies with lock mass. The Byonic software 4.2.10 (protein metric) was used for peptide identification. The search parameter was set as: precursor mass tolerance ~20 ppm, fragment mass tolerance ~20 ppm, maximum missed cleavages ~5, and cysteine carbamidomethylation (+57.021464), oxidation (+15.994915), dimedone adduct (+138.068068) and methionine oxidation (+15.994915) with target protein database. Orf1 (45 µM) was incubated in the presence of 2.5 mM glycine and 5 mM **4** or **7** at 37 °C for 1 hour in 92.2% (v/v) $H_2^{18}O$. The reaction solution was exchanged to a buffer solution containing 25 mM $NH_4HCO_3$ with 5 µg µl⁻¹ trypsin. After 24 h incubation at 37 °C, the samples (2 µg µl⁻¹) were injected and detected by LC-ESI-MS on an Orbitrap Fusion mass spectrometer (Thermo Fisher Scientific, San Jose, CA) equipped with EASY-nLC 1200 system (Thermo, San Jose, CA, US) and EASY-spray source (Thermo, San Jose, CA, US). The digestion solution was injected (5 µl) at 1 µl/min flow rate on to easy column (C18, 0.075 mm×150 mm, ID 3 µm; Thermo Scientific) Chromatographic separation was using 0.1% formic acid in water as mobile phase A and 0.1% formic acid in 80% acetonitrile as mobile phase B operated at 300 nl/min flow rate. Briefly, the gradient employed was 2% buffer B at 2 min to 40% buffer B at 40 min. Full-scan MS condition: mass range m/z 375-1800 (AGC target 5E5) with lock mass, resolution 60,000 at m/z 200, and maximum injection time of 50 ms. Target m/z were isolated for CID with NCE35 and maximum injection time of 100 ms. Electrospray voltage was maintained at 1.8 kV and capillary temperature was set at 275 °C.

## NMR experiments

[1]H- and [13]C-NMR spectra of compounds **1** and **8** were recorded at 500 and 150 MHz, respectively, using a Bruker AMX-500 spectrometer. One- and two- dimensional experiments (correlation spectroscopy (COSY), heteronuclear multi quantum correlation-total correlation spectroscopy (HMQC-TOCSY), and heteronuclear multiple bond correlation (HMBC)) were performed at ambient temperature in $D_2O$ (Supplementary Figs. 2, 4; Supplementary Table 1, 2). [1]H- (600 MHz) and [13]C-NMR (150 MHz) spectra of compounds **4, 7, 9** and **10** were recorded by Bruker Avance 600 MHz in $D_2O$ (Supplementary Figs. 20–27), and their chemical shifts were tabulated in Supplementary Table 5–8. The structures of 10- and 12-carbarmoyl **4, 9**, and **10** were determined by NMR experiments.

## Enzyme assay of glycine oxidation

The quantity of glycine oxidation was estimated according to the production of glyoxylate[42]. Glyoxylate was reacted with 2,4-dinitrophenylhydrazine forming 2,4-dinitrophenylhydrazone derivatives which can be detected at 445 nm. The quantity of the derivatives was interpolated against the standard curve of 0.078-1.25 mM glyoxylate. The enzyme (20 μM) was incubated with addition of 0.2–3 mM glycine in 100 mM sodium phosphate pH 8. The samples (150 μl) were withdrawn at 0, 30 and 60 seconds and mixed with 75 μl 1 mM 2,4-dinitrophenylhydrazine in 1 M HCl. After 10 mins incubation at 37 °C, the reactions were added with 525 μl 0.6 N NaOH for 10 mins incubation at room temperature. The absorbance at 445 nm was measured by using a DU® 800 UV/Visible spectrophotometer. The relative activity of wild-type and mutants were performed with 20 μM enzyme and 2 mM glycine in 100 mM sodium phosphate pH 8 at 37 °C. The data were analyzed by GraphPad Prism using non-linear regression model for fitting the Michaelis-Menten equation.

## Enzyme Kinetic assay of iminoacetyl-ST-F production

The production quantity of **8** was estimated according to the consumption of **7**. The standard curve was made on the basis of the peak area of 0–500 μM ST-F at EIC ~ 503.2. Each reaction contained 2.5 μM enzyme and 25–500 μM ST-F in 100 mM sodium phosphate pH 8 at 37 °C over 10 mins. The reactions were quenched by chloroform. After centrifugation, supernatants were injected in to a TSKgel® Amide-80 column (5 μm, 4.6 mm×250 mm) formic acid. mounted on a Water Alliance 2695 HPLC equipped with a Xevo TQ-S micro triple quadrupole mass spectrometer. The gradient employed was 65% (v/v) acetonitrile for 5 min and 65–20% (v/v) acetonitrile for 25 min. Both acetonitrile and water contained 0.1% (v/v) formic acid. The data were analyzed by GraphPad Prism using non-linear regression model for fitting the Michaelis-Menten equation.

## Measurement of the production of hydrogen peroxide

The hydrogen peroxide production during enzyme reactions was measured using the Pierce™ Quantitative Peroxide Assay Kit [https://www.thermofisher.com/order/catalog/product/tw/en/23280]. Each reaction contained 25 μM enzyme, 250 μM glycine in 100 mM sodium phosphate pH 8 buffer at 30 °C for 3 mins. A sample of 20 μl was added with 200 μl working reagent at room temperature for 20 mins. The absorbance at 595 nm was measured using a SpectraMax® M5 microplate reader.

## Binding affinity determination of Orf1 with ST-F

The binding affinities of ST-F versus Orf1 wild-type or mutants were estimated by using MicroCal iTC200 isothermal titration calorimetry (Malvern Panalytical). The exothermic heat pulse was recorded when 2 μl of ST-F (1–3 mM) was injected to a 200 μl protein solution (0.1 mM) containing 25 mM HEPES pH 7.5 at 25 °C. The differential binding curve was fitted with a standard single-site binding model of Origin 7 (Supplementary Fig. 28).

## Disc diffusion assay

The *E. coli* strains (BL21 (DE3) and ATCC 25922) were grown at 37 °C overnight and spread on the LB agar plate. Five μl of each individual compound (10 mM) were loaded onto paper discs and placed on the bacterial lawn of plates. All the plates were incubated overnight at 37 °C to observe the zone of inhibition.

## Reporting summary

Further information on research design is available in the Nature Portfolio Reporting Summary linked to this article.

## Data availability

The data associated with this study are available within the article, supplementary information and Source Data file. The mass spectrometry data have been deposited to the ProteomeXChange Consortium via the PRIDE partner repository with the identifier PXD041103 (username:reviewer_pxd041103@ebi.ac.uk, password:WBKFJVIV), PXD041104 (username:reviewer_pxd041104@ebi.ac.uk, password:ad9bPAJK) and PXD041105 (username:reviewer_pxd041105@ebi.ac.uk, password:3-nYMJGKU). All the coordinates and structure factors have been deposited in the Protein Data Bank with accession codes 7XYE, 7XQA, 7XXD, 7XXC, 7XXM, 7XX0, 7XXR, 7XYL, 7XXP, 7Y0X, 8GRI, 7YPU, and 7YPV.Source data are provided with this paper. All other data are available from the corresponding author on request. Source data are provided with this paper.

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

## Acknowledgements

This work was supported by funds from the Ministry of Science and Technology (MOST), Taiwan (MOST-108-2113-M-001-021-MY3, MOST-110-0210-01-22-02 and MOST-111-2113-M-001-031-MY3) and Academia Sinica (AS-KPQ-109-BioMed, 109-0210-01-18-02 and AS-IA-109-L06). This research was also supported by a JSPS KAKENHI grant for Scientific Research on Innovative Areas 16H06445 (C.M and Y.H), by Grant-in-Aid for Transformative Research Areas 22H05122 (C.M.), and by the JSPS A3 Foresight Program. We thank the experimental facility and the technical services provided by the Synchrotron Radiation Protein Crystallography Core Facility of the National Core Facility for Biopharmaceuticals, Ministry of Science and Technology, and the National Synchrotron Radiation Research Center (NSRRC), a national user facility funded by the Ministry of Science and Technology, Taiwan (R.O.C.). We thank NSRRC for beam time allocations at beam lines TPS-05A, TPS-07A, and TLS-15A. Thank Yi-Chun Liu (NSRRC) for the diffraction-weighted dose estimation. Mass spectrometry analysis were performed by Mass Core Facility of Genomics Research Center, Academia Sinica, Taiwan. We thanked the NMR center of Genomics Research Center for technical support. Thanks also go to Instrumentation Center at NTHU for Bruker ElEXSYS E580 EPR analysis. We thank T. Ito and T. Hibi for discussion on the biochemical analysis of Orf1.

## Author contributions

T.L.L. and Y.H. initiated the study. T.L.L. and Y.H. designed experiments. Y.L.W., C.Y.C. and C.L.C. expressed proteins. Y.L.W. determined protein structures, prepared substrates, designed mutants, and performed enzymatic assays and mass spectrometric analysis. Y.L.W., N.S.H. and K.H.L performed EPR and ITC experiments. I.W.L. and C.F.C performed NMR analysis. Z.C.W. prepared knockout strains. Y.O., T.D., C.M. and Y.H. provided consultations with respect to the BD-12 producing strain and engaged in discussions. T.L.L., Y.H. and Y.W.L. wrote the manuscript.

## Competing interests

The authors declare no competing interests.
