## [Peer Review File · Nature Communications]

N-Formimidoylation/-iminoacetylation modification in aminoglycosides requires FAD-dependent and ligand-protein NOS bridge dual chemistryREVIEWER COMMENTS

Reviewer #1 (Remarks to the Author):

The authors have studied the structure and mechanism of an unusual flavoenzyme that catalyses formation of the antibiotic BD-12 by ligation of an N-formimidoyl moiety and glycinotrigin. They employed X-ray crystallography to obtain snapshots along the pathway and intriguingly observed a covalent conjugate between an active center cysteine and the reaction intermediate in the form of an iminoglycine N-O-S cysteine. They sketch out a mechanism and confirm their hypothesis by the finding that the cysteine is essential for catalysis, any mutant protein is enzymatically inactive. Also, the enzyme is (almost) inactive under reducing conditions further supporting their claim. Overall, the finding is fascinating and the structural biology has been executed very well, the unbiased omit maps clearly prove the chemical states as reported. One key issue this referee would like to address is the following. The authors assign the Cys281 to exist in the monooxidized form (sulfenic acid) under turnover conditions (dimedone probe). Would formation of the sulfenic acid be kinetically competent for the overall reaction (relative to k_{cat})? Is formation of the sulfenic acid an requirement of the reaction or could the oxime-like intermediate directly react with the thiol form (non-oxidized) cysteine to afford the NOS linkage?

Reviewer #2 (Remarks to the Author):

This manuscript describes a very nice story about Orf1, a flavoenzyme identified in *Streptomyces* strains. This work is supported by a huge number of results obtained by a combination of bioinformatic analysis, mass spec, enzymology and crystallography data. The enzyme Orf1 coordinates two reaction centers to carry out a sequence of reactions that add a glycine-derived N-formimidoyl group to glycinotrigin, which belongs to the streptothricins (STs) family of anti-microbials.

There are few issues that should be addressed:

- in general, introduction is a bit confusing: the enzymatic system is complex and it should be better described also in relation to the homologous enzymes cited in the text. Also, the classification of FPMO categories is not so clear: is Orf I or III? What are exactly the enzymes functioning with one-center or two-center mechanisms?
 - Fig. 1: it is a complicated figure that it should be more precisely cited in the introduction with all a,b,c panels (see issue above). Panel b: the right-side part shows two reactions (decarboxylation and oxidation) starting from the same molecule and leading to products 5 or 6: are these two alternative reactions that may occur? It would be helpful for the reader to indicate "glycylthricin" under the drawn structure because all structures in panel a are quite complicated. Though obvious, also "glycine" can be added.
 - Fig. 2c: in the text (page 7), residues 54 and 55 are mentioned as important for glycine binding but they are not shown in the figure.
 - Page 7 and Fig. 2f: the cavity originally supposed to be the glycylthricin binding site is not small, is it filled by water molecules? Where is located with respect to the true glycylthricin cavity and to Cys281? What could be the role of this cavity?
 - Fig. 3: panels d-g should be commented at page 9. Although the explanation of the role of the various residues is given in figure legend, this aspect should be mentioned also in the text.
- Moreover, the scheme shown in Fig. 1c (blue squared) is an anticipation of the results obtained in the work: it could be moved in Fig. 3. The zoomed inset in Fig. 3b is not necessary.
- Is the monooxygenase character of Orf1 supported by spectrophotometric studies? Was the peroxy-flavin detected?
 - The sentence at the end of the conclusion is not clear.
 - Extended Data Fig. 2b: why does Orf1 incubated with glycine become purple?

Minor issues:

- Fig. 1 legend: acetylation (not acetatylation)
- check measurement unit of beamsize in Suppl Material file at page 3 which lost the correct format
- check spelling of crystallized at page 6
- check spelling of stabilized and adduct in the sentence between page 9 and page 10.
- check spelling of sarcosine in Extended Data Table 2

Reviewer #3 (Remarks to the Author):

The authors comprehensively studied the N-formimidoylation/-imidoacetylation modification of glycinothricin family compounds mediated by FAD-dependent oxidoreductase Orf1, which oxidize glycine into imine and imine-oxygen adduct in its FAD-binding reactive center and further install the glycine imine species on the terminal amino moiety of the substrates in the other reactive center via a special "Cys281-S-O-N-glycine imine" intermediate. In vivo and in vitro characterization and isotope labeling of one of the co-substrates, glycine, were conducted to illustrate the catalysis of Orf1, and apo/complex crystal structures of wild type or variants were resolved to demonstrate the dual reactive center and the unique N-O-S bridge; the physiological role of N-formimidoylation/-imidoacetylation was also studied, which was assumed as an evasion strategy to the drug resistance of competing species.

Both the chemistry and the physiological role of Orf1 are interesting, and great efforts were made to illustrate them. However, the analysis of experimental results and expression of idea are not fluent, making the manuscript bear low readability; the processing of key data was quite rough (Fig 2-4, see below) that it's hard to comprehend the authors' experiments and analysis; confusing statements and linguistic errors can be found throughout the manuscript and some conclusion is not fully supported by current data. These major problems should be addressed before further consideration of publication.

1. Where is the O atom from in the N-O-S linkage? Based on the proposed mechanism, it derived from O₂ mediated by FAD-dependent oxidation of glycine, but experimental confirmation is deficient. ¹⁸O₂ labeling experiment may be useful to clarify this issue. Related with this, in the catalytic mechanisms of Orf1, the authors' "two scissile bond" theory inferred that N-O bond breakage lead to decarboxylation of products (2 and 5), while S-O bond breakage lead to N-iminoacetyl products (3, 6, and 8). H²¹⁸O labeling proved the latter (Fig 4h), however, the former lacks similar evidence.

2. The authors believed that reactive species formed at the first reaction center is "tunneled" to the second one. This is an important feature in the catalysis of Orf1. How can the authors rule out a re-enter mechanism? The coupled assays (ThiO/R342A and C281S/R342A) are not sufficient to prove this, for the reactive species may be water labile. Does point mutation of residue(s) in the "tunnel" block the passage of glycine imine and thus abolish the formation of the N-O-S adduct and the end products? Experimental validation is required.

3. "N-formimidoylation is ...a biochemical strategy that the producing strain evolved to reinvigorate its armory to counter the orthodox drug-resistance mechanism leveraged by competing species." This conclusion cannot be well supported by current experimental result. Actually, only one aminoglycoside-modifying enzyme, SttE, was tested in this study, and no competing species (only E. coli harboring sttE) was tested. Besides, the authors should discuss why N-formimidoyl/-iminoacetyl products evade the inactivation of SttE based on the structural information, which is important and can direct future drug development.

4. "Orf1" is not characteristic enough to be a recognizable tag in database, please give a special name to this unique enzyme.

5. P1, Abstract, engaging (engaged), forming (to form)

6. P3, "On the other hand, a marked contrast in antibacterial effects and stability between the FDA-approved β -lactam antibiotic imipenem (N-formimidoylated) and thienamycin 38 bodes well the modification given that N-formimidoyl fortimicin A (dactimicin) is resistant to AAC3 and with low nephrotoxicity." This sentence is confusing.

7. P7, the details of reaction center, including H bond, π - π stacking, distance between atoms, are required to be labeled in a zoomed in illustration.

8. P7, "as it seems accessible through a narrow entrance for the glycyI side chain of glycyIthricine to an oxidized glycine" Can't figure out the "narrow entrance" or its accessibility in this figure.

9. P8, "the reactive product generated from glycine oxidation in the FAD-mediated reaction is safeguarded and swiftly transferred to the glycyIthricin binding site allowing". As mentioned above, how did the authors prove this proceed? There's no evidence to rule out the re-enter mode of oxidated glycine.

10. Fig4 note, (c) Mass spectra of iminoacetyl-glycyIthricin 6 produced in the Orf1-mediated reactions in the presence of glycyIthricin 3 and... should be "glycyIthricin 4"

11. Fig4 b and c can be combined into one graph that display the MS of both compound 5 and 6.

12. (e) was assigned as (b) by mistake.

13. Fig4h, when 7 was used as substrate, a 2 amu increment in mass spectra of C281 containing fragment of Orf1 can be detected, while the same increment was not detected when 4 as substrate. This result evidenced the proposed non-decarboxylative pathway (Fig1c). However, it conflicts with the enzyme assay of Orf1 in Fig4a(ii), in which N-iminoacetylated product 6 was also obviously detected.

14. Fig2 please label FAD and residues in different color pattern.

15. Fig2a, the direction of rotation seems wrong.

16. Fig2f, it's difficult to recognize the glycyIthricin binding site, please highlight in the figure.

17. Extended figure 1, MS data are ambiguous. Why the abundance of $[M+H]^+$ is minor in the spectra? Standard 1 and 2 are required, or test again with purified 1 and 2.

18. Structure of compounds should be listed in Extended Fig 5-7.

19. Supporting information P4, "Constructing an orfV-deficient mutantz". This gene was not mentioned in the manuscript. Please check.

20. Supporting information P5, "Isothermal Titration Calorimetry Analysis (ITC) of ST-F 6 for its binding affinity with Orf1 and mutants". ST-F 6 should be "ST-F 7".

Reviewers' comments:

The authors have studied the structure and mechanism of an unusual flavoenzyme that catalyses formation of the antibiotic BD-12 by ligation of an *N*-formimidoyl moiety and glycinothricin. They employed X-ray crystallography to obtain snapshots along the pathway and intriguingly observed a covalent conjugate between an active center cysteine and the reaction intermediate in the form of an iminoglycine N-O-S cysteine. They sketch out a mechanism and confirm their hypothesis by the finding that the cysteine is essential for catalysis, any mutant protein is enzymatically inactive. Also, the enzyme is (almost) inactive under reducing conditions further supporting their claim. Overall, the finding is fascinating and the structural biology has been executed very well, the unbiased omit maps clearly prove the chemical states as reported. One key issue this referee would like to address is the following. The authors assign the Cys281 to exist in the monooxidized form (sulfenic acid) under turnover conditions (dimedone probe). Would formation of the sulfenic acid be kinetically competent for the overall reaction (relative to k_{cat})? Is formation of the sulfenic acid an requirement of the reaction or could the oxime-like intermediate directly react with the thiol form (non-oxidized) cysteine to afford the NOS linkage?

Reply: We thank reviewer #1 for his/her general positive comments. With regard to the issue "Would formation of the sulfenic acid be kinetically competent for the overall reaction (relative to k_{cat})?" our answer is positive. The estimation for the conversion ratio for the sulfhydryl of Cys281 to its sulfenic acid in the presence of glycine is as high as 0.54 (54%) in 30 seconds (Fig. S16) (and there are no observable modifications occurring at other cysteine residues in Orf1), indicating that the sulfenic acid modification is intrinsic and kinetically competent. As for the related

concern “Is formation of the sulfenic acid a requirement of the reaction or could the oxime-like intermediate directly react with the thiol form (non-oxidized) cysteine to afford the NOS linkage?”, the formation of the sulfenic acid is indeed required for the reaction. Additionally, we have compared the turnover rates for the *N*-formimidoyl product formation between two reactions with addition of the same quantity of Orf1 and glycine-pretreated Orf1 (almost all sulfhydryl of Cys281 converted to sulfenic acid), whereby they are almost identical, supporting that the sulfenic acid is kinetically competent and required. The multiple reactions catalyzed by Orf1 are intriguing and complicated, which should deserve a detailed, comprehensive kinetic study separately and be published in a specialized biochemical journal. On the other hand, one literally cannot rule out that the oxime-like intermediate may directly react with the thiol form (non-oxidized) cysteine, while as mentioned in the text the likelihood that two nucleophiles react to form a covalent bond is low. Two other possible intermediates oxaziridin and cyclic peroxide glycine shown in the Figure 1, that may respectively react with oxidized and non-oxidized cysteine to form the NOS linkage, however, were not detected in any circumstances.

Reviewer #2 (Remarks to the Author):

This manuscript describes a very nice story about Orf1, a flavoenzyme identified in *Streptomyces* strains. This work is supported by a huge number of results obtained by a combination of bioinformatic analysis, mass spec, enzymology and crystallography data. The enzyme Orf1 coordinates two reaction centers to carry out a sequence of reactions that add a glycine-derived *N*-formimidoyl group to glycinotrigin, which belongs to the streptothricins (STs) family of anti-microbials.

There are few issues that should be addressed:

- in general, introduction is a bit confusing: the enzymatic system is complex and it should be better described also in relation to the homologous enzymes cited in the text. Also, the classification of FPMO categories is not so clear: is Orf I or III? What are exactly the enzymes functioning with one-center or two-center mechanisms?

Reply: We like to thank reviewer #2 for his/her general positive comments. For the comment of “introduction is a bit confusing”, we have rewritten and reorganized the section to some extent. Regarding the classification, glycine oxidase (ThiO), an enzyme essential for the biosynthesis of thiamine, is the most similar one to Orf1 in terms of sequence similarity and biochemical reaction. ThiO has been characterized as a class III flavoprotein oxidase (FPOX) oxidizing glycine to glycine imine with

concomitant formation of H₂O₂ in one single reaction center. In contrast, Orf1 is characterized herein as a class III flavoprotein monooxygenase (FPMO), that oxidizes glycine in one reaction center and the oxidized product is channeled to a second reaction center where it is coupled with glycinothricin to form *N*-formimidoylated glycinothricin with the mediation of activated Cys281.

- Fig. 1: it is a complicated figure that it should be more precisely cited in the introduction with all a,b,c panels (see issue above). Panel b: the right-side part shows two reactions (decarboxylation and oxidation) starting from the same molecule and leading to products 5 or 6: are these two alternative reactions that may occur? It would be helpful for the reader to indicate "glycylthricin" under the drawn structure because all structures in panel a are quite complicated. Though obvious, also "glycine" can be added.

Reply: As referred above, we have followed the reviewer's suggestion adding in more information and/or rephrasing description in clarity including Fig. 1b in Introduction.

- Fig. 2c: in the text (page 7), residues 54 and 55 are mentioned as important for glycine binding but they are not shown in the figure.

Reply: For this unclarity, we have rephrased the sentence as "Except for G55 and H284, the residues A54, M57, R368 and R342 that surround the glycine/FAD reaction center are conserved between Orf1 and ThiO in the secondary and ternary sequence level" in the text.

- Page 7 and Fig. 2f: the cavity originally supposed to be the glycylothricin binding site is not small, is it filled by water molecules? Where is located with respect to the true glycylothricin cavity and to Cys281? What could be the role of this cavity?

Reply: The originally supposed glycylothricin binding site is filled with water molecules. It is located at one end of the FAD/glycine binding site connected through a bottleneck, wherefore it may serve as a gate controlling the entrance of glycine/oxygen (Extended data Fig. 11a). The true glycylothricin binding site comprising Cys281 (Fig. 2f) is located at the other end of the FAD/glycine binding site and separated by β -strands including β 11- β 15 (Fig. 2c, 2f and 2j). This spatial arrangement expedites the reaction along the assembly line in an orderly and controllable manner.

- Fig. 3: panels d-g should be commented at page 9. Although the explanation of the role of the various residues is given in figure legend, this aspect should be mentioned also in the text. Moreover, the scheme shown in Fig. 1c (blue squared) is an anticipation of the results obtained in the work: it could be moved in Fig. 3. The zoomed inset in Fig. 3b is not necessary.

Reply: We followed the reviewer's suggestion describing the observations of Fig. 3 d-g in the figure legend and text. The C281-adduct and the glycylothricin-containing complexes were superpositioned showing that the distance between the amine group of glycylothricin and α -carbon of the iminoacetate adduct is 3.1 Å in a general H-bond range with an approaching trajectory in a Burgi-Dunitz angle of 99.7° (Fig. 3de). We next overlaid complexes WT-glycylothricin **4** (green), E312A-glycylothricin **4** (cyan) and C281-iminoglycine adduct (respectively colored green, cyan and grey in Fig. 3f) revealing that E312 undergoes a substrate-induced conformational change, in which upon addition decarboxylation should facilitate β -elimination. We further superpositioned complexes WT-glycylothricin **4** (green), WT-4-aminobutylthricin **10** (magenta), F316-ST-F **7** (cyan) and C281-iminoglycine adduct (respectively colored green, magenta, cyan and grey in Fig. 3g) identifying considerable displacements of the median chain γ -aminobutyl and the long chain β -lysine away from that of glycylothricin likely as a result of a geographic effect.

- Is the monooxygenase character of Orf1 supported by spectrophotometric studies? Was the peroxy-flavin detected?

Reply: Oxidase and monooxygenase that distinguish from each other lie on two facts: Namely, only does the former produce hydrogen peroxide, while a transient peroxy-flavin is formed in the latter. In the present study, Orf1 is capable of generating few detectable leaky hydrogen peroxide. In contrast, Orf1 displays a characteristic signal of peroxy-flavin at 395 nm by stop-flow PDA spectrophotometric analysis (Fig. S17), underlining that Orf1 is a member of the monooxygenase superfamily.

- The sentence at the end of the conclusion is not clear.

Reply: We have rephrased the sentence as "The NOS-mediated chemistry catalyzed by Orf1 should not be the sole case, for reactions/enzymes that borrow similar chemistry ought to be more out there if one tracks down the geometric signature uncovered herein at the gene or protein level (Fig. 1c)"

- Extended Data Fig. 2b: why does Orf1 incubated with glycine become purple?

Reply: Color change for reactions catalyzed by an FAD-containing enzyme (Orf1) is typical and characteristic. The fact is FAD_{ox} (yellow) is reduced with concomitant oxidation of substrates (glycine), where FAD_{ox} (yellow), half reduced FAD (semiquinone/hydroquinone, blue or red) and FAD_{red} (colorless). The purple suggests the reaction solution is likely a mixture of all possible components, particularly C4-peroxide-flavin (purple blue).

Minor issues:

- Fig. 1 legend: acetylation (not acetatylation)
- check measurement unit of beam size in Suppl Material file at page 3 which lost the correct format
- check spelling of crystallized at page 6
- check spelling of stabilized and adduct in the sentence between page 9 and page 10.
- check spelling of sarcosine in Extended Data Table 2

Reply: We thank the reviewer for spotting all these typos and errors, which are all fixed now.

Reviewer #3 (Remarks to the Author):

The authors comprehensively studied the *N*-formimidoylation/-imidoacetylation modification of glycinothricin family compounds mediated by FAD-dependent oxidoreductase Orf1, which oxidize glycine into imine and imine-oxygen adduct in its FAD-binding reactive center and further install the glycine imine species on the terminal amino moiety of the substrates in the other reactive center via a special “Cys281-S-O-N-glycine imine” intermediate. In vivo and in vitro characterization and isotope labeling of one of the co-substrates, glycine, were conducted to illustrate the catalysis of Orf1, and apo/complex crystal structures of wild type or variants were resolved to demonstrate the dual reactive center and the unique N-O-S bridge; the physiological role of *N*-formimidoylation/-imidoacetylation was also studied, which was assumed as an evasion strategy to the drug resistance of competing species.

Both the chemistry and the physiological role of Orf1 are interesting, and great efforts were made to illustrate them. However, the analysis of experimental results

and expression of idea are not fluent, making the manuscript bear low readability; the processing of key data was quite rough (Fig 2-4, see below) that it's hard to comprehend the authors' experiments and analysis; confusing statements and linguistic errors can be found throughout the manuscript and some conclusion is not fully supported by current data. These major problems should be addressed before further consideration of publication.

Reply: We like to thank reviewer #3 for his/her general positive comments. In this revised version, we have addressed all the concerns raised by reviewer #3 and improved its readability to some extent.

1. Where is the O atom from in the N-O-S linkage? Based on the proposed mechanism, it derived from O₂ mediated by FAD-dependent oxidation of glycine, but experimental confirmation is deficient. ¹⁸O₂ labeling experiment may be useful to clarify this issue. Related with this, in the catalytic mechanisms of Orf1, the authors' "two scissile bond" theory inferred that N-O bond breakage lead to decarboxylation of products (2 and 5), while S-O bond breakage lead to *N*-iminoacetyl products (3, 6, and 8). H₂¹⁸O labeling proved the latter (Fig 4h), however, the former lacks similar evidence.

Reply: It has been well established that cysteine's reactivity makes it uniquely susceptible to oxidation by reactive oxygen species (ROS) in the cell ^{49,52}. One-electron oxidations of cysteine to radical species can occur, as well as two-electron oxidations to form disulfide bonds or various advanced oxidization species. In the present study, although few Orf1 is capable of producing ROS hydrogen peroxide at the FAD-mediated reaction center ²⁶, where molecular oxygen is reduced by receiving electrons from reduced FAD to form reactive oxidative intermediates including escaped or meaningful leaky hydrogen peroxide. Our protein mass spec analysis revealed that Cys281 is the only residue out of total 12 Cys residues sensitive to hydrogen peroxide. Hydrogen peroxide is diffusible but likely deliberately delivered through a specific route to reach Cys281 (see below). Our EPR analysis that shows barely detectable radical signal suggests the appearance of sulfenic acid, sulfinic acid and/or sulfonic acid at Cys281 (detected by mass spec) follows typical two-electron oxidation, where the sulfenic group of Cys281 is the active form easily undergoing molecular-oxygen-mediated oxidation reactions to sulfinic acid or sulfonic acid when substrates are depleted (glycine and/or glycinotrigin). Moreover, the Orf1-mediated enzymatic reaction cannot proceed in the presence of fresh reducing agents (e.g., DTT), highlighting the oxidation activation to sulfenic acid is critical, commensurate

with the so-called PTM (protein posttranslational modification). It has long been appreciated that the oxygen atoms on sulfenic, sulfinic and sulfonic acids are of molecular oxygen origin. We do not challenge this well-established fact, while two possible routes emerge in this study given two products *N*-formimidoyl and *N*-iminoacetyl glycinethricin formed in the HOS-Cys281 mediated reaction as shown in Fig. 1c. In the former, HOS-Cys281 can be regenerated; in the latter, HOS-Cys281 could be regenerated in two ways, that is, re-oxidization of reduced Cys281 (SH-Cys281) by hydrogen peroxide (derived from molecular oxygen) or through water-mediated bond cleavage reaction (which was not reported previously). To examine the possibility of the latter, one can perform the reaction in oxygen-18 labeled water, which literally proves the likelihood that unexpectedly is the major route, while the oxygen-18 molecular oxygen experiment seems duplicate providing no intrinsic merit.

2. The authors believed that reactive species formed at the first reaction center is “tunneled” to the second one. This is an important feature in the catalysis of Orf1. How can the authors rule out a re-enter mechanism? The coupled assays (ThiO/R342A and C281S/R342A) are not sufficient to prove this, for the reactive species may be water labile. Does point mutation of residue(s) in the "tunnel" block the passage of glycine imine and thus abolish the formation of the N-O-S adduct and the end products? Experimental validation is required.

Reply: We thank the reviewer’s suggestion. We took advantage of CAVER 3.0 to predict possible tunnels that branch out from the FAD/glycine binding site (Extended data Fig. 11a). Four possible tunnels around the FAD/glycine reaction center were predicted one for the oxidized glycine channeling and three for entrance of glycine. Five residues that constitute the tunnel and would block the intermediate tunneling if mutated were individually made (L124D, L124E, A278V, A280S and H419Y). Except L124E (insoluble), all the mutants L124D, A278V and H419Y lost the *N*-iminoacetylation activity unable to form compound **8** despite that they are still active for glycine oxidation (Extended Data Fig. 11b). Both software-added prediction and biochemical experiments validate the one-way tunneling effect.

3. “*N*-formimidoylation is ...a biochemical strategy that the producing strain evolved to reinvigorate its armory to counter the orthodox drug-resistance mechanism leveraged by competing species.” This conclusion cannot be well supported by

current experimental result. Actually, only one aminoglycoside-modifying enzyme, SttE, was tested in this study, and no competing species (only *E. coli* harboring sttE) was tested. Besides, the authors should discuss why *N*-formimidoyl/-iminoacetyl products evade the inactivation of SttE based on the structural information, which is important and can direct future drug development.

Reply: We thank the reviewer's suggestion. We added more information in the text. The bifunctional aminoglycoside modifying enzyme AAC(6')-Ie-APH(6'')-Ia⁵⁶ inactivates glycylothricin by *N*-acetylation to some extent in *in vitro* assay (Fig. S15), while the *N*-formimidoylated glycylothricin is not sensitive to this given modifying enzyme. It is likely due to the reduced nucleophilicity of the sp² *N*-formimidoyl group of **5** where the imine is in resonance with the neighboring α -amino group of **4**.

4. "Orf1" is not characteristic enough to be a recognizable tag in database, please give a special name to this unique enzyme.

Reply: We thank the reviewer's suggestion. Orf1 is functionally termed as *N*-formimidoyl/-iminoacetyl- synthase, which has been added in Introduction.

5. P1, Abstract, engaging (engaged), forming (to form)

Reply: We have fixed the errors.

6. P3, "On the other hand, a marked contrast in antibacterial effects and stability between the FDA-approved β -lactam antibiotic imipenem (*N*-formimidoylated) and thienamycin 38 bodes well the modification given that *N*-formimidoyl fortimicin A (dactimicin) is resistant to AAC3 and with low nephrotoxicity." This sentence is confusing.

Reply: We thank the reviewer's comment. The sentence has been rephrased to "An encouraging example is that FDA-approved β -lactam antibiotic imipenem that is *N*-formimidoylated thienamycin³⁶, demonstrating superior efficacy to the latter for both drug resistance and stability. Similarly, *N*-formimidoyl fortimicin A (dactimicin) is insusceptible to AAC3 as well as has lower nephrotoxicity^{37,38}".

7. P7, the details of reaction center, including H bond, π - π stacking, distance between atoms, are required to be labeled in a zoomed in illustration.

Reply: The distance between the NOS-adduct and Phe316 has been labeled (Fig. 3e) in the revised version. The distances are within the range of 4.1-5.1 Å.

8. P7, "as it seems accessible through a narrow entrance for the glycyI side chain of glycyIthricine to an oxidized glycine" Can't figure out the "narrow entrance" or its accessibility in this figure.

Reply: To make it clearer, we have added Extended Data Fig. 11a, where the narrow entrance is pointed out.

9. P8, "the reactive product generated from glycine oxidation in the FAD-mediated reaction is safeguarded and swiftly transferred to the glycyIthricin binding site allowing". As mentioned above, how did the authors prove this proceed? There's no evidence to rule out the re-enter mode of oxidated glycine.

Reply: Please refer to question 2.

10. Fig4 note, (c) Mass spectra of iminoacetyl-glycyIthricin 6 produced in the Orf1-mediated reactions in the presence of glycyIthricin 3 and... should be "glycyIthricin 4"

Reply: We have corrected the mistake.

11. Fig4 b and c can be combined into one graph that display the MS of both compound 5 and 6.

Reply: Figures 4b and 4c represent spectra for two different products **5** and **6**, respectively. We kept as they are to avoid any potential confusion.

12. (e) was assigned as (b) by mistake.

Reply: We have corrected the typo.

13. Fig4h, when 7 was used as substrate, a 2 amu increment in mass spectra of C281 containing fragment of Orf1 can be detected, while the same increment was not detected when 4 as substrate. This result evidenced the proposed non-decarboxylative pathway (Fig1c). However, it conflicts with the enzyme assay of Orf1 in Fig4a(ii), in which N-iminoacetylated product 6 was also obviously detected.

Reply: We added a new sentence in the figure legend "The reaction condition is the same as that for the experiments shown in Fig. S13c(ii) and Extended Data Fig. 7e(ii). Two major products are *N*-formimidoyl-glycinthricin **5** and *N*-iminoacetyl-ST-F **8**, respectively."

14. Fig2 please label FAD and residues in different color pattern.

Reply: We have changed the color of FAD in Fig. 2.

15. Fig2a, the direction of rotation seems wrong.

Reply: We have fixed the direction of the rotation in Fig. 2a.

16. Fig2f, it's difficult to recognize the glycylothricin binding site, please highlight in the figure.

Reply: We have regenerated Fig. 2f, where the binding surface is shown on the left side of Cys281.

17. Extended figure 1, MS data are ambiguous. Why the abundance of [M+H]⁺ is minor in the spectra? Standard 1 and 2 are required, or test again with purified 1 and 2.

Reply: The MS data shown in Extended fig. 1 was reaction directly subjecting the culture medium broth into mass spectrometer for analysis. The mass information for purified compound 1 and 2 is provided in supplementary information (Fig. S13a, b).

18. Structure of compounds should be listed in Extended Fig 5-7.

Reply: We have followed the reviewer's suggestion adding chemical structures of amikacin and sisomicin in Extended Data Fig. 5 and 6. We also assigned compound's numbers in the Extended Data Fig. 7.

19. Supporting information P4, "Constructing an *orfV*-deficient mutantz". This gene was not mentioned in the manuscript. Please check.

Reply: We have added the information of *orfV* in the section of Methods.

20. Supporting information P5, "Isothermal Titration Calorimetry Analysis (ITC) of ST-F 6 for its binding affinity with Orf1 and mutants". ST-F 6 should be "ST-F 7".

Reply: We have fixed the error.

We, once again, thank all reviewers for their thorough and comprehensive review and valuable comments.

REVIEWERS' COMMENTS

Reviewer #2 (Remarks to the Author):

The authors addressed all issues raised in the previous review and replied to clarify the aspects that were not clear.

Only the last sentence of the Conclusion section is still a bit foggy and contorted: I caught the general meaning, but how could one track the geometry (i.e. the active site architecture, I guess) at the gene or protein level? They could simply state that the sequence fingerprint characterizing the active associated to this type of chemistry may guide in finding more enzymes featuring the same mechanism.

Reviewer #3 (Remarks to the Author):

I believe the authors have responded well to the reviewers comments. I favor publication in Nature Communications.

REVIEWERS' COMMENTS

Reviewer #2 (Remarks to the Author):

The authors addressed all issues raised in the previous review and replied to clarify the aspects that were not clear.

Only the last sentence of the Conclusion section is still a bit foggy and contorted: I caught the general meaning, but how could one track the geometry (i.e. the active site architecture, I guess) at the gene or protein level? They could simply state that the sequence fingerprint characterizing the active associated to this type of chemistry may guide in finding more enzymes featuring the same mechanism.

Reviewer #3 (Remarks to the Author):

I believe the authors have responded well to the reviewers comments. I favor publication in Nature Communications.